# Design and structural validation of peptide–drug conjugate ligands of the kappa-opioid receptor

Edin Muratspahić[1,14,17], Kristine Deibler [2,15,17], Jianming Han[3,17], Nataša Tomašević[1], Kirtikumar B. Jadhav[4], Aina-Leonor Olivé-Marti[5], Nadine Hochrainer[5], Roland Hellinger[1], Johannes Koehbach[6,16], Jonathan F. Fay [7], Mohammad Homaidur Rahman[8], Lamees Hegazy[8], Timothy W. Craven[2], Balazs R. Varga[3,9], Gaurav Bhardwaj[2], Kevin Appourchaux [3,9], Susruta Majumdar [3,9], Markus Muttenthaler[4,10], Parisa Hosseinzadeh [11], David J. Craik [6], Mariana Spetea [5], Tao Che [3,9] ✉, David Baker [2,12,13] ✉ & Christian W. Gruber [1] ✉

Despite the increasing number of GPCR structures and recent advances in peptide design, the development of efficient technologies allowing rational design of high-affinity peptide ligands for single GPCRs remains an unmet challenge. Here, we develop a computational approach for designing conjugates of lariat-shaped macrocyclized peptides and a small molecule opioid ligand. We demonstrate its feasibility by discovering chemical scaffolds for the kappa-opioid receptor (KOR) with desired pharmacological activities. The designed De Novo Cyclic Peptide (DNCP)-β-naloxamine (NalA) exhibit in vitro potent mixed KOR agonism/mu-opioid receptor (MOR) antagonism, nanomolar binding affinity, selectivity, and efficacy bias at KOR. Proof-of-concept in vivo efficacy studies demonstrate that DNCP-β-NalA(1) induces a potent KOR-mediated antinociception in male mice. The high-resolution cryo-EM structure (2.6 Å) of the DNCP-β-NalA–KOR–Gi1 complex and molecular dynamics simulations are harnessed to validate the computational design model. This reveals a network of residues in ECL2/3 and TM6/7 controlling the intrinsic efficacy of KOR. In general, our computational de novo platform overcomes extensive lead optimization encountered in ultra-large library docking and virtual small molecule screening campaigns and offers innovation for GPCR ligand discovery. This may drive the development of next-generation therapeutics for medical applications such as pain conditions.

G protein-coupled receptors (GPCRs) are critical therapeutic targets for analgesics, antihistamines, neuroleptics, and many cardiovascular drugs[1,2]. Small molecules are a common therapeutic modality for targeting GPCRs[2] due to their low cost, high stability, lipophilicity, and oral bioavailability; however, they often have limited target selectivity and are associated with off-target effects and adverse clinical events[3,4]. A prominent example is the ongoing and rapidly evolving global opioid crisis accompanied by substantial opioid-related morbidity and mortality[5,6]. Prescribed opioid analgesics including fentanyl, morphine and their derivatives, that act primarily via the mu-opioid receptor

(MOR), have numerous and serious side effects[7]. The kappa-opioid receptor (KOR) is an attractive alternative of therapeutic value for the development of non-addictive pain relievers[8,9]. Nevertheless, while KOR is a promising target, its activation can result in other undesired effects, including psychotomimesis, sedation and dysphoria[10], which has limited the clinical development of KOR agonists as analgesic drugs[9,11]. Intensive research over the past decade has thus focused on the design of molecular scaffolds targeting KOR with improved side effect profiles. For instance, functionally selective small molecule KOR ligands, partial KOR agonists and mixed KOR/MOR agonists, exhibited fewer side effects in preclinical settings[9,11,12]. However, the utility of functionally selective (biased) ligands of opioid receptors remains controversial[13,14]. Hence, there is an unmet need to discover KOR ligands that display unique pharmacological properties and diminished unwanted effects. Previously, this has been attempted using high-throughput small molecule library screening[15], virtual small molecule screening[16] or natural product discovery[17], but with limited translational success. In parallel, advances in computational design have enabled the design of structured cyclic peptides and small proteins with high affinity and selectivity for a variety of targets[18–23]. Despite this progress, there have been few successes in structure-guided peptide design for GPCRs, mainly due to the lack of high-resolution active-state GPCR structures[24] and the recessed ligand binding pockets of many small molecule sensing GPCRs which can be difficult for peptides to access.

We reason that the advantages of small molecule ligands in penetrating deeply into GPCR binding clefts and of peptides in making more extensive target interactions for specificity could be combined by computational design of peptide–small molecule conjugates. Utilizing a small molecule ligand extended by a peptide moiety allows access simultaneously to orthosteric and alternative binding sites of GPCRs, enabling the modulation of different active states of the receptor[25], which could endow such ligands with unique pharmacological properties.

Here, we exploit the crystal structure of KOR bound to the dual KOR/delta-opioid receptor (DOR) epoxymorphinan opioid agonist MP1104[8] and use the Rosetta protein and peptide design software to computationally design peptide–small molecule conjugates targeting KOR. Utilizing the high affinity interaction of MP1104[26] with KOR as an anchor to initiate the design calculations, we seek to computationally design thioether cyclic peptides that interact with ECL2 and ECL3 of the receptor[27].

## Results

### Computational design of KOR-targeted peptide–small molecule conjugates

We developed a computational design approach using Rosetta peptide design methods[18,21] to design high affinity cyclic peptide–small molecule drug conjugate ligands targeting KOR (Fig. 1). The recently published structure of human KOR bound to small molecule MP1104 (PDB: 6B73)[8], reveals multiple interactions between MP1104 and KOR (e.g., T111[2.56], F114[2.59], Q115[2.60], Y139[3.33] and V230[5.43]) that contribute to binding (Supplementary Fig. 1). We reasoned that a small molecule ligand resembling MP1104 could provide affinity for KOR, and that selectivity and efficacy could be modulated by a conjugated cyclic peptide interacting with the extracellular loops. To enable chemical synthesis of this conjugate, we decided to use lariat peptides cyclized through a sidechain and the N-terminus, with the C-terminus thus available for covalent modification. The cyclic motif embedded within the peptide sequence reduces flexibility and thus reduces the entropy loss upon binding which can increase binding affinity and enhance stability[4,28]. Based on the size and shape of the binding pocket, we chose to employ the cyclic component of the lariat 5-6 residues closed by thioether macrocyclization linking a Cys side chain and the N-terminus (Fig. 1b).

We started from a variant of MP1104 lacking the iodobenzamide group−N-cyclopropylmethyl- epoxy morphinan−bound to the KOR structure in the same orientation as MP1104. This MP1104 derivative has a free amine, which can be conjugated to the C-terminus of a peptide lariat. We first modeled two amino acid residues extending off the free amino group of the MP1104 derivative, and extensively sampled their backbone torsion angles. Next to the free amino group, we placed a D-phenylalanine to mimic the MP1104 iodobenzamide group, and at the second position we sampled all 20 amino acids (excluding glycine and cysteine) in L and D forms, aiming for interactions with the extracellular loops of the receptor (Supplementary Fig. 1c, d). Over all combinations of backbone conformations and amino acid possibilities, we chose four solutions (i.e., dipeptides D-Phe-L-Thr; D-Phe-L-Ser; D-Phe-L-Gln and D-Phe-L-Ala) with the lowest Rosetta binding energy for KOR that were next used to graft 5- and 6-mers thioether cyclized peptides onto.

We next generated a library of thioether cyclized peptides to graft onto the di-peptide-morphinan models. We used Rosetta to generate a large set of cyclic thioether peptide scaffolds with 5 or 6 amino acids using backbone generation and clustering protocols previously employed to generate backbone cyclized peptides (Supplementary Data 1)[21]. The thioether scaffolds were fused to the generated tails by coordinate transformations to form amide bonds between the C-termini of the thioether cyclic peptide backbones and the N-termini of the tail segments (Supplementary Data 2). We performed iterative sequence design rounds on the peptide backbone to optimize interactions with the KOR pocket using the Rosetta FastDesign protocol (Supplementary Data 3)[18,20,21]. After each round of design, the computed free energy of binding (ΔΔG), the shape complementarity, and the interface area were assessed, and a 90th percentile cut-off on all three properties was used to filter the designs down to 50 structures (Fig. 1, Supplementary Fig. 2)[20]. The designs were further filtered based on interactions with the extracellular loops (ECL2 and/or ECL3) (Fig. 2a) and the diversity of the sequences, shape of the cyclic backbone, and extracellular loop interactions across ECL2/3 (Fig. 2b). Six thioether cyclized hexamer conjugates were selected for synthesis (Fig. 2c).

### Chemical synthesis of the designed ligands and determination of their pharmacodynamic properties at KOR

We first synthesized de novo linear (DNLP) and de novo cyclic peptides (DNCP) (Fig. 2c and Supplementary Fig. 3a) to test if they can bind to the orthosteric site of KOR on their own to validate the conjugate design. Hence, DNLPs (11–16) were synthesized by Fmoc-solid phase peptide synthesis (Fmoc-SPPS) and purified by RP-HPLC (Supplementary Fig. 4, Supplementary Tables 1 and 2). The DNCPs (21–26) were also assembled by Fmoc-SPPS, followed by a three-step procedure consisting of N-terminal on-resin bromoacetylation, chemoselective Mmt deprotection of Cys[6], and a final on-resin thioether cyclization to form the 6-residue macrocycle (Supplementary Fig. 5, Supplementary Tables 1 and 2). Then DNLPs (11–16) and DNCPs (21–26) were assayed in one-point radioligand binding assays using membrane preparations from HEK293 cells stably expressing the mouse KOR. None of the tested peptides was able to displace the orthosteric antagonist [³H] diprenorphine ([³H]DPN) from KOR at a concentration of 10 μM (Supplementary Fig. 6).

For synthesis of the peptide−drug conjugates, we chose β-NalA, distinguished by a less rigid morphinan structure and featuring an N-17 allyl group over the N-cyclopropylmethyl group of the MP1104 derivative (i.e., N-cyclopropylmethyl-epoxy morphinan), which was utilized during computational design. This decision was guided by the closely resembling core structure, along with the benefits of β-NalA's simpler synthesis and a more flexible morphinan ring, thereby allowing the peptide region to bind the ECL2 region of KOR more tightly[29]. Conjugation of the DNCPs (31–36) (Supplementary Fig. 7 and

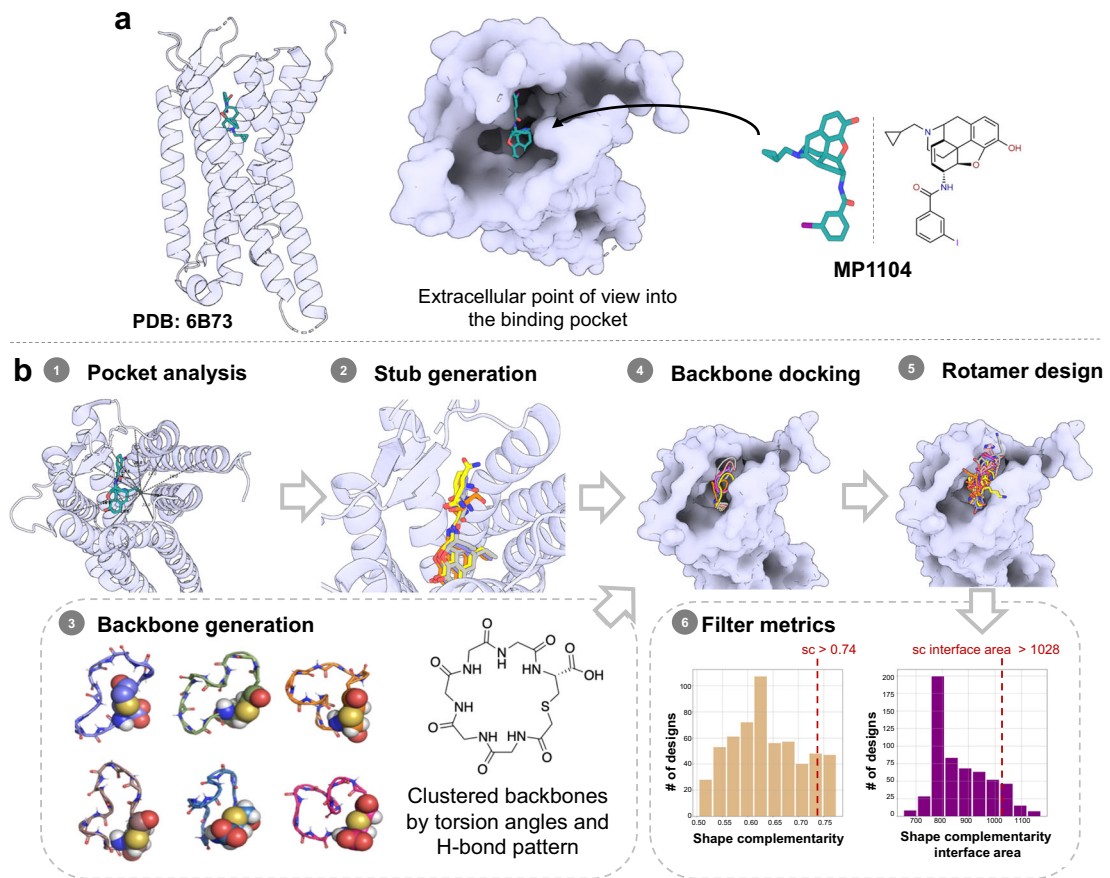

**Fig. 1 | Strategy for the computational design of thioether macrocyclized peptide–small molecule conjugates targeting KOR. a** 3D human KOR structure with small molecule agonist MP1104 was used as the starting template (PDB: 6B73). **b** Workflows for computational peptide–small molecule conjugate design: (1) Measurement of the pocket area (934 Å3) narrowed down the size of macrocycles to focus on 5- and 6-mer cyclic peptides. (2) Generation of small molecules with two additional amino acids, which were sampled and scored for optimal dimer sequence (select dipeptide modified small molecules: Gray: CVV-D-Phe-Thr; Yellow: CVV-D-Phe-Gln; Orange: CVV-D-Phe-Ser; CVV corresponds to N-cyclopropylmethyl-epoxy morphinan small molecule stub). (3) Generation of a comprehensive library of 5- and 6-mer thioether cyclized peptides clustered via torsion angle and hydrogen bond pattern. (4) Docked structure of thioether macrocyclized hexamers through coordinate-guided transformation of the backbone C-termini to the generated anchor N-termini. (5) Rotamer design to optimize the interface interactions of the backbones. (6) Design filtering based on shape complementarity and interface area as representative examples for interface metrics; dashed red line represents 90th percentile cut-off values.

Supplementary Table 2) to β-NalA was achieved by coupling the C-terminal carboxylic acid (COOH) of the DNCPs to the primary amino group (NH₂) of β-NalA (Fig. 2c and Supplementary Fig. 3). We obtained four DNCP-β-NalA conjugates (**1**–**4**) (Supplementary Fig. 8 and Supplementary Table 2); the conjugation of DNCP (**35**) and DNCP (**36**) was unsuccessful (Supplementary Fig. 9). DNCP-β-NalA conjugates (**1**–**4**) were pharmacologically characterized via radioligand binding and functional cAMP assays to investigate their affinity, potency and efficacy at the mouse KOR (Fig. 3a, b and Supplementary Table 3). The DNCP-β-NalA conjugates exhibited affinities in the low nanomolar range (Supplementary Table 3), with DNCP-β-NalA(**1**) being the strongest binder at KOR with a $K_i$ value of 3.9 nM (Fig. 3a and Supplementary Table 3), as compared to β-NalA, which has an affinity $K_i$ of 72 nM. All four DNCP-β-NalA conjugates were full agonists at KOR ($E_{max}$ = 89–101%), while β-NalA alone was only a partial agonist with an $EC_{50}$ of 130 nM and $E_{max}$ of 61% (Fig. 3b and Supplementary Table 3). The most potent conjugates were DNCP-β-NalA(**1**) and (**4**) with $EC_{50}$ values of 2.0 nM and 1.0 nM, respectively (Fig. 3b and Supplementary Table 3). DNCP-β-NalA(**2**) and DNCP-β-NalA(**3**) showed agonist activities at KOR with $EC_{50}$ values of 7.5 and 14 nM in cAMP assay, respectively, while their $K_i$ values were 31 and 24 nM in radioligand binding assay, respectively (Supplementary Table 3). Receptor reserve may account for this discrepancy between potency and affinity values of

DNCP-β-NalA(**2**) and DNCP-β-NalA(**4**) which can be observed in functional GPCR assays with opioid receptors[30]. The higher potency and efficacy of the conjugates at KOR compared to β-NalA can be attributed to interactions of the macrocycles with the ECL2 region as described later.

## Pharmacological profiling of DNCP-β-NalA(1) for receptor subtype selectivity and functional bias

We pursued more detailed pharmacological characterization of DNCP-β-NalA(**1**) to obtain information regarding the affinity and activation at MOR and DOR, as well as the potency/efficacy of G proteins vs. β-arrestins. Given the sequence differences between human and mouse KOR (Supplementary Fig. 10) we evaluated the in vitro functional activity of DNCP-β-NalA(**1**) at the human KOR in the [³⁵S]GTPγS binding assay using CHO cell membrane preparations stably expressing human KOR. DNCP-β-NalA(**1**) fully activated human KOR ($EC_{50}$ = 5.5 nM; $E_{max}$ = 83%) compared to the partial agonist β-NalA ($EC_{50}$ = 150 nM; $E_{max}$ = 22%) and the reference KOR agonists U69,593 and dynorphin (dyn) A₁₋₁₃ (Fig. 3c and Supplementary Table 4), which is in agreement with the activation of the mouse KOR as determined by cAMP quantification (Fig. 3b).

Pathway-selective KOR ligands offer great potential not only as molecular probes to dissect receptor pharmacology but also as

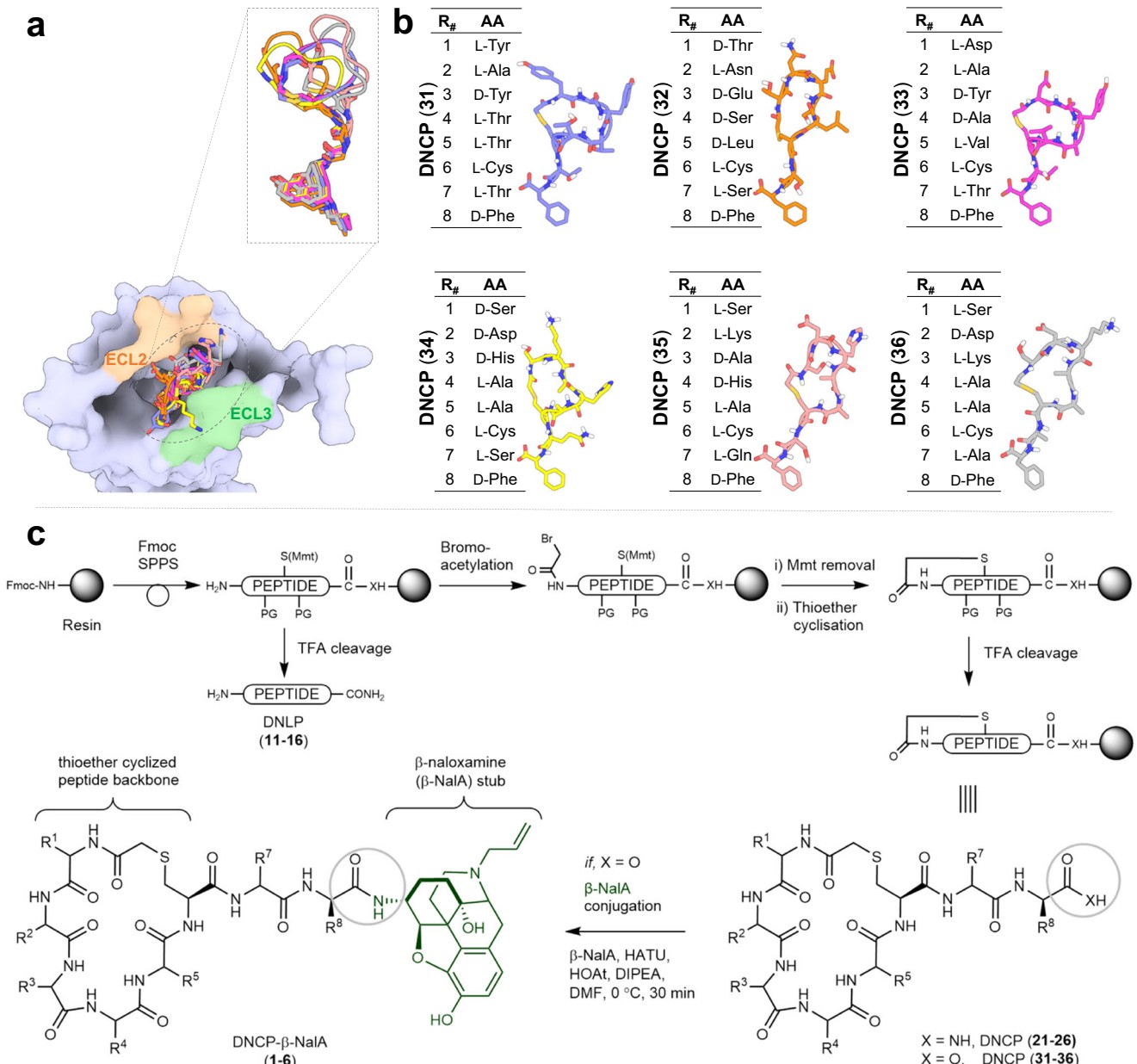

**Fig. 2 | Overview of the selected peptide macrocycle designs and the synthetic strategy to produce the peptide–small molecule conjugates and controls.**
**a** Peptides have interactions with ECL2 and/or ECL3 of KOR with a subset overlay of the peptide backbone designs (aligned by the small molecule fragment anchor) showing shape diversity in the pocket. **b** Final selection of peptide sequences. Instead of focusing on multiple sequences for a single promising backbone, we sought to select designs across diverse shapes and sequences for experimental testing. **c** Synthetic scheme depicting solid phase synthesis of the de novo linear

peptides (DNLP) (**11–16**) and the de novo cyclic peptides (DNCP) (**21–26** and **31–36**) and the solution phase conjugation reaction with β-naloxamine (β-NalA) to generate the DNCP-β-NalA (**1–6**) conjugates. R# indicates a side chain of the respective amino acid. PG denotes protecting groups. Rink amide resin was used for synthesis of DNLP (**11–16**) and DNCP (**21–26**), whereas Fmoc-D-Phe preloaded Wang resin was used for DNCP (**31–36**) synthesis. The amino acids indicated by R# in Fig. 2b correspond to the identical side chain represented by R# in Fig. 2c.

---

potential non-addictive next-generation analgesics. In this context, β-arrestins have been linked to severe adverse effects, thereby highlighting the potential of G protein-biased ligands to design pain relievers with improved side effect profiles[9,31,32]. Thus, we explored the ability of DNCP-β-NalA(**1**) to recruit β-arrestin-2 at KOR in a bioluminescence resonance energy transfer (BRET) assay (Fig. 3d, i, j). In contrast to reference ligands dynorphin (dyn) $A_{1-13}$ and U69,593 (full agonists of β-arrestin-2, with an $EC_{50}$ of 31 and 500 nM, respectively) DNCP-β-NalA(**1**) only partially recruited β-arrestin-2 with an $E_{max}$ of 41% and an $EC_{50}$ of 22 nM (Fig. 3d and Supplementary Table 3). Similarly, β-NalA recruited β-arrestin-2 at KOR with an $E_{max}$ and an $EC_{50}$ of 51% and

15 nM, respectively. DNCP-β-NalA(**1**) also partially recruited β-arrestin-1 at KOR with an $E_{max}$ of 30% and $EC_{50}$ of 28 nM (Fig. 3i, j, Supplementary Fig. 11a and Supplementary Table 3). Hence, DNCP-β-NalA(**1**) demonstrated impaired β-arrestin recruitment, established to be associated with reduced side effects[31,33,34]. We then profiled DNCP-β-NalA(**1**) in the TRUPATH assay to interrogate their $G\alpha_{i/o}$ coupling preferences Fig. 3e, i, j, Supplementary Fig. 11b–d, Supplementary Tables 5 and 6)[35]. The TRUPATH screening platform has been developed as an alternative to cAMP second messenger assays to minimize signal overamplification[35]. Intriguingly, in contrast to U50,488 and MP1104, but similar to the mixed-action KOR agonist pentazocine, DNCP-β-NalA(**1**) exhibited

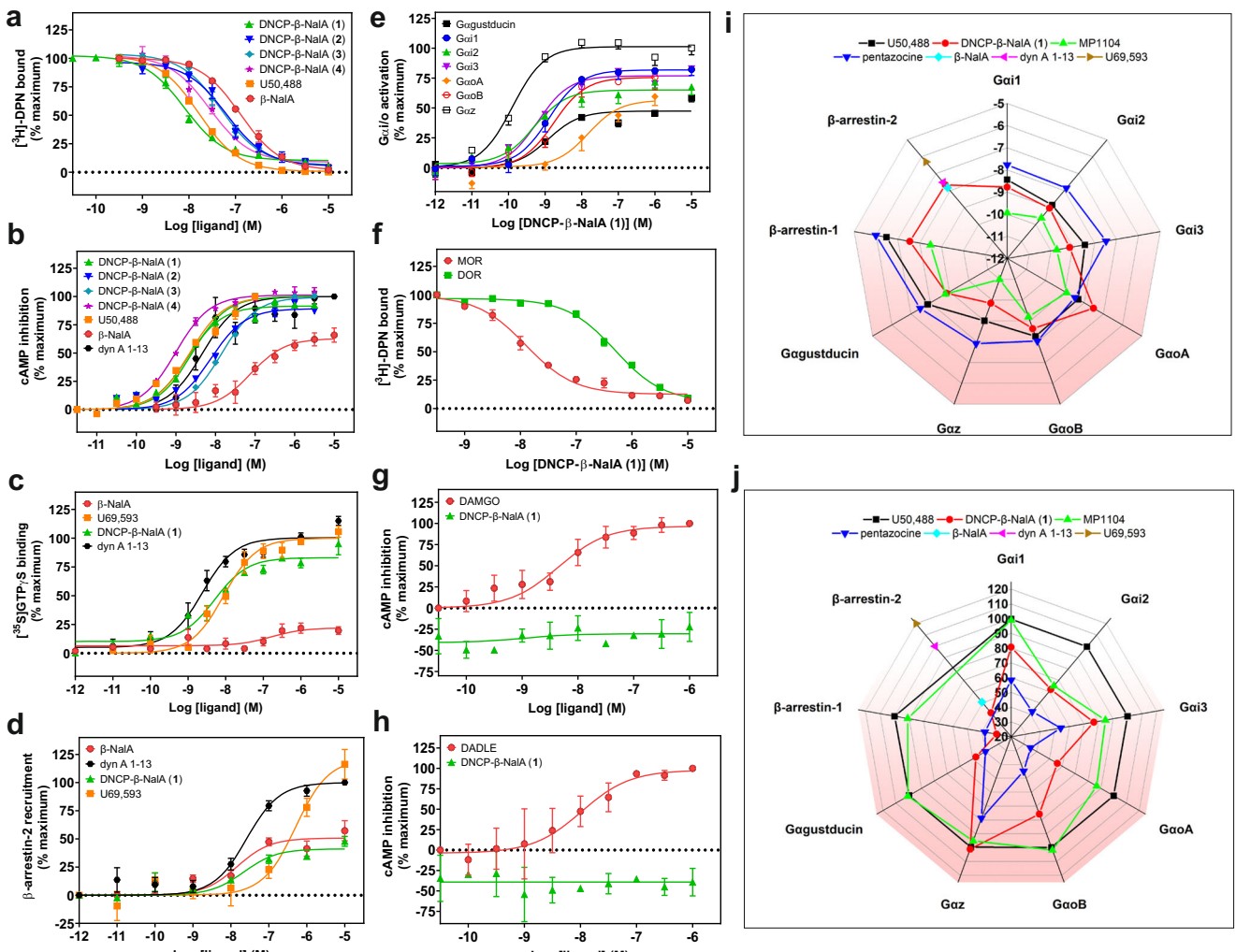

**Fig. 3 | In vitro receptor pharmacology of peptide–small molecule conjugates.**
**a**, **b** Radioligand binding ($n = 3$) and functional cAMP assays ($n = 3$–4) of DNCP-β-NalA conjugates (**1**–**4**) were performed on HEK293T cell membranes stably expressing mouse KOR. Binding (**a**) was measured by displacing 1 nM of [$^3$H]DPN whereas cAMP inhibition (**b**) was monitored after treatment with indicated concentrations of conjugates. U50,488 and β-NalA were positive controls. Final concentration of 10 μM of forskolin was used to stimulate cAMP production (Supplementary Table 3). **c** Concentration-dependent stimulation of [$^{35}$S]GTPγS binding by the most potent DNCP-β-NalA(**1**) ($n = 3$), β-NalA ($n = 3$), U69,593 ($n = 3$) and dyn A$_{1-13}$ ($n = 4$) in human KOR expressing CHO cell membranes (Supplementary Table 4). **d** β-arrestin-2 recruitment assay of DNCP-β-NalA(**1**), β-NalA and dynorphin (dyn) A$_{1-13}$ was done in HEK293T cells transiently expressing mouse KOR-EGFP and β-arrestin-2-nano-luciferase ($n = 3$–6) (Supplementary Table 3). **e** α-

Subtype screening of DNCP-β-NalA(**1**) at the mouse KOR in the TRUPATH assay ($n = 8$) (Supplementary Tables 5 and 6). **f** Selectivity of DNCP-β-NalA(**1**) was determined in a radioligand binding assay using HEK293T cell membrane preparations stably expressing mouse MOR and DOR and 1 nM of [$^3$H]DPN, respectively ($n = 3$). **g**, **h** Gα$_i$-mediated cAMP inhibition of DNCP-β-NalA(**1**) at the mouse MOR (**g**) and DOR (**h**) was measured in stable HEK293T cells using DAMGO and DADLE as reference ligands, respectively ($n = 3$). **i**, **j** Spider plots from TRUPATH ($n = 8$ for each Gα), β-arrestin-1 ($n = 4$) and β-arrestin-2 ($n = 3$–6) recruitment assays represent potency (log $EC_{50}$) (**i**) and normalized efficacy (**j**) of DNCP-β-NalA(**1**), U50,488, MP1104, pentazocine, β-NalA and dyn A$_{1-13}$. Data were normalized to full KOR agonists U50,488, U69,593 or dyn A$_{1-13}$, full MOR agonist DAMGO and full DOR agonist DADLE. All data are presented as mean values ± s.e.m. Source data are provided as a Source Data file.

KOR partial agonism at Gα$_{i2}$, Gα$_{i3}$, Gα$_{oA}$, Gα$_{oB}$ and Gα$_{gastducin}$ subtypes with $E_{max}$ values ranging from 48% to 77%, whereas it elicited full agonism at Gα$_{i1}$ and Gα$_z$ with $E_{max}$ values of 81% and 101%, respectively (Fig. 3i, j, Supplementary Table 6). The highest potency in the picomolar range was observed at Gα$_z$ subtype for DNCP-β-NalA(**1**), U50,488 and MP1104, while pentazocine revealed the least variation in potency and efficacy across the transducerome (Fig. 3i, j, Supplementary Fig. 11b–d, Supplementary Tables 5 and 6).

Next, we determined the opioid receptor subtype selectivity profile of DNCP-β-NalA(**1**) in radioligand binding assays with membrane preparations from HEK293 cells stably expressing the mouse MOR and DOR, and CHO cells stably expressing human nociceptin (NOP) receptor. Herein, DNCP-β-NalA(**1**) bound to mouse MOR and DOR with $K_i$ values of 5.4 and 318 nM, respectively, supporting an ~80-

fold selectivity for KOR over DOR (Fig. 3f) whereas it bound to the human NOP with a $K_i$ value of ~1.3 μM, thus having an ~330-fold selectivity for KOR over NOP receptor (Supplementary Fig. 12). In the functional cAMP assay, DNCP-β-NalA(**1**) was inactive at both mouse MOR and DOR up to 10 μM (Fig. 3g, h). This was confirmed at the human MOR and DOR in the [$^{35}$S]GTPγS binding assay (Supplementary Fig. 13 and Supplementary Table 4). We next determined the mechanism of antagonism of DNCP-β-NalA(**1**) by measuring adenylyl cyclase-mediated cAMP inhibition and [$^{35}$S]GTPγS binding at mouse and human MOR, respectively, using Schild regression analysis. The MOR expressed in HEK293 and CHO cells was activated by DAMGO in the absence and presence of increasing concentrations of DNCP-β-NalA(**1**). We observed a rightward shift of the concentration-response curves of DAMGO in cAMP (Supplementary Fig. 14a) and [$^{35}$S]GTPγS

binding assays (Supplementary Fig. 14b). Schild analysis of DNCP-β-NalA(**1**) exhibited linear regression slopes of 0.9 and 1.5 (Supplementary Fig. 14c, d) and pA2 values of 9.1 and 7.9 in cAMP and [$^{35}$S]GTPγS binding assays, respectively, which corresponds to an average functional affinity of 0.8 and 13 nM, respectively, thus demonstrating the competitive antagonism of the DNCP-β-NalA(**1**) at MOR.

### Antinociceptive efficacy and lack of typical KOR-mediated side effects of designed peptide–small molecule conjugate

To further support in vitro findings of DNCP-β-NalA(**1**), we confirmed its serum stability (>95% remaining intact after 48 h; Supplementary Fig. 15) and then examined its antinociception in vivo after subcutaneous (s.c.) administration to male mice. Antinociceptive effects were evaluated in two mouse models of pain, the formalin test and Complete Freund's Adjuvant (CFA)-induced inflammatory hyperalgesia. In the formalin test, s.c. administration of DNCP-β-NalA(**1**) produced a dose-dependent reduction in the pain behavior of formalin-injected male mice with significant effects at doses of 1.9 and 3.8 µmol kg$^{-1}$ (Fig. 4a). The prototypical KOR agonist U50,488 also produced a dose-dependent decrease in the nociceptive response with a significant effect at all tested doses (Fig. 4b). The calculated antinociceptive $ED_{50}$ values in the formalin test of DNCP-β-NalA(**1**) and U50,488 were 1.89 µmol kg$^{-1}$ (95% confidence limits, 95% CL: 0.97–3.66) and 1.64 µmol kg$^{-1}$ (95% CL: 0.77–3.53), respectively, revealing that DNCP-β-NalA(**1**) was equipotent to U50,488. To evaluate the involvement of KOR in DNCP-β-NalA(**1**)-induced antinociception, the effect of the selective KOR antagonist nor-binaltorphimine (nor-BNI) was tested. Pretreatment of male mice with nor-BNI (13.6 µmol kg$^{-1}$, s.c.) significantly reversed the antinociceptive response of DNCP-β-NalA(**1**), confirming a KOR-dependent antinociceptive effect of DNCP-β-NalA(**1**) in the formalin test (Fig. 4c).

Next, we investigated the antinociceptive efficacy of DNCP-β-NalA(**1**) after s.c. administration in male mice with CFA-induced inflammatory hyperalgesia. Male mice received CFA to the dorsal side of the right hindpaw, and hyperalgesia was evidenced by a significant reduction at 72 h post-inoculation versus pre-innoculation (P < 0.001, paired t-test) in paw withdrawal thresholds to thermal stimulation assessed with the Hargreaves test. Male mice were treated s.c. with saline, and different doses of DNCP-β-NalA(**1**), or U50,488, and tested for thermal sensitivity (Fig. 4d, e, respectively). DNCP-β-NalA(**1**) produced time- and dose-dependent increase in the inflamed paw withdrawal latencies. Compared to saline-treated mice, DNCP-β-NalA(**1**) significantly reduced thermal sensitivity at doses of 0.8 and 1.9 µmol kg$^{-1}$ (Fig. 4d). Notable was the fast onset of the antihyperalgesic effect of DNCP-β-NalA(**1**) with a peak effect at 15 min followed by a rapid decline, with thermal nociceptive thresholds returning to basal values at 2 h after drug administration.

Administration of U50,488 also caused a dose-dependent attenuation in pain behavior of male mice with CFA-induced inflammatory hyperalgesia (Fig. 4e). Doses of 0.6 and 2.1 mg kg$^{-1}$ of U50,488 significantly increased paw withdrawal latencies from 15 min to 1 h, and from 15 min to 6 h, respectively, with a peak antinociceptive effect at 30 min. Although DNCP-β-NalA(**1**) had a shorter duration of the antinociceptive effect than U50,488, it showed comparable antinociceptive efficacy at the highest tested dose in attenuating the pain response in male mice with CFA-induced inflammatory hyperalgesia. We also demonstrated that the antinociceptive effect of DNCP-β-NalA(**1**) (1.9 mg kg$^{-1}$) was reversed by pretreatment with the nor-BNI (13.6 µmol kg$^{-1}$, s.c.), indicating that the KOR is involved in DNCP-β-NalA(**1**) in vivo agonist activity (Fig. 4f). Altogether, these data show that DNCP-β-NalA(**1**) efficiently reversed thermal hyperalgesia in male mice with CFA-induced inflammatory pain acting through the KOR, with a fast onset of action.

We then explored the potential anti-inflammatory effect of DNCP-β-NalA(**1**) after s.c. administration in male mice by measuring paw thickness, 60 min after formalin injection into the right hind paw. DNCP-β-NalA(**1**) significantly reduced paw thickness of the formalin-injected paw at doses of 1.9 and 3.8 µmol kg$^{-1}$, by 12 ± 2% and 17 ± 3%, respectively, compared to saline-treated mice (Fig. 4g). By contrast, U50,488 did not affect paw oedema formation at any of the tested doses when compared to the saline group (Fig. 4h). The DNCP-β-NalA(**1**) inhibitory effect on the formalin-induced paw inflammation was significantly reversed by pre-treatment with nor-BNI (13.6 µmol kg$^{-1}$, s.c.), indicating the contribution of KOR to the anti-inflammatory effect of DNCP-β-NalA(**1**) (Fig. 4i).

Finally, we investigated the behavioral effects of DNCP-β-NalA(**1**) on motor coordination and the potential to induce sedation in male mice using the rotarod test (Fig. 4j). Male mice were administered the highest antinociceptive effective dose in the formalin test, i.e., 3.8 µmol kg$^{-1}$ of DNCP-β-NalA(**1**) and 5.4 µmol kg$^{-1}$ of U50,488. The rotarod test was also performed in male mice receiving a dose of 7.6 µmol kg$^{-1}$ of DNCP-β-NalA(**1**). While U50,488 caused a significant deficit in the rotarod performance at 30 min in comparison with saline mice, DNCP-β-NalA(**1**) produced no changes in the motor behavior of male mice with no significant alterations in rotarod latencies at any time point and tested doses. These data demonstrate that DNCP-β-NalA(**1**) produces significant and potent antinociception in the formalin test and CFA-induced inflammatory hyperalgesia, along with anti-inflammatory effects, without KOR-mediated liability of motor dysfunction/sedation after s.c. administration in male mice.

### Structural validation of peptide–small molecule design

High-resolution structures are instrumental for evaluating the accuracy of computationally designed molecules. We thus determined the cryo-EM structure of human KOR bound to DNCP-β-NalA(**1**) and Gα$_{i1}$/Gβ$_1$/Gγ2 heterotrimer at the nominal resolution of 2.6 Å (Fig. 5a, Supplementary Fig. 16 and Supplementary Table 7). This high resolution enables the unambiguous modeling of KOR and G protein heterotrimer (Supplementary Fig. 16 and Supplementary Fig. 17). The overall structure of the KOR-G$_{i1}$ protein heterotrimer is similar to the MOR-G$_{i1}$ structure[36], likely due to stabilization by the same G protein subtype (Supplementary Fig. 18a, b). The major difference is in the Gα helix where the N-terminal α-helix (αN) in KOR-G$_{i1}$ has a 3 Å displacement and α$_1$ has a 2 Å displacement compared to that in MOR-G$_{i1}$ structure. KOR adopts an active-state conformation with the hallmark of 10 Å outward movement of transmembrane helix 6 (TM6) compared to the inactive-state KOR[37] (Supplementary Fig. 18c).

DNCP-β-NalA(**1**) occupies two major binding pockets of KOR. One is the orthosteric site by the small molecule portion (Fig. 5b), which is also the canonical binding site for "classical" KOR ligands; the other pocket is in the extracellular region formed by ECL2, ECL3, TM6, and TM7 (Fig. 5c), which is occupied by the peptide moiety. The small molecule portion of DNCP-β-NalA(**1**) (bottom half) adopts a conformation similar to MP1104, as well as other typical KOR agonists including U50,488 and pentazocine (Supplementary Fig. 19). Comparison of the computationally designed model DNCP-β-NalA(**1**) with the experimental structure shows a conserved binding mode of the D-Phe at position 1. The initial goal of the design was to form interactions with both the ECL2 and ECL3. We are still seeing these interactions but in an altered manner: in the designed model the peptide was predicted to have a D-Tyr (DNCP1_design D-Tyr–R$_3$) interaction with ECL2 and a Tyr (DNCP1_design Tyr–R$_1$) interaction with ECL3. We found that the residue D-Tyr (R$_3$) interacted with the ECL3 versus the predicted interaction with ECL2 (Supplementary Fig. 20). Further structural information of the other computationally designed peptides whose ECL2/3 interactions are similar to those of computational DNCP-β-NalA(**1**) (Supplementary Fig. 21) would elucidate the peptides' ability to induce an altered conformation upon binding. To explore the stability and molecular interactions of the peptide component in the extracellular pocket, we conducted a total of four individual molecular

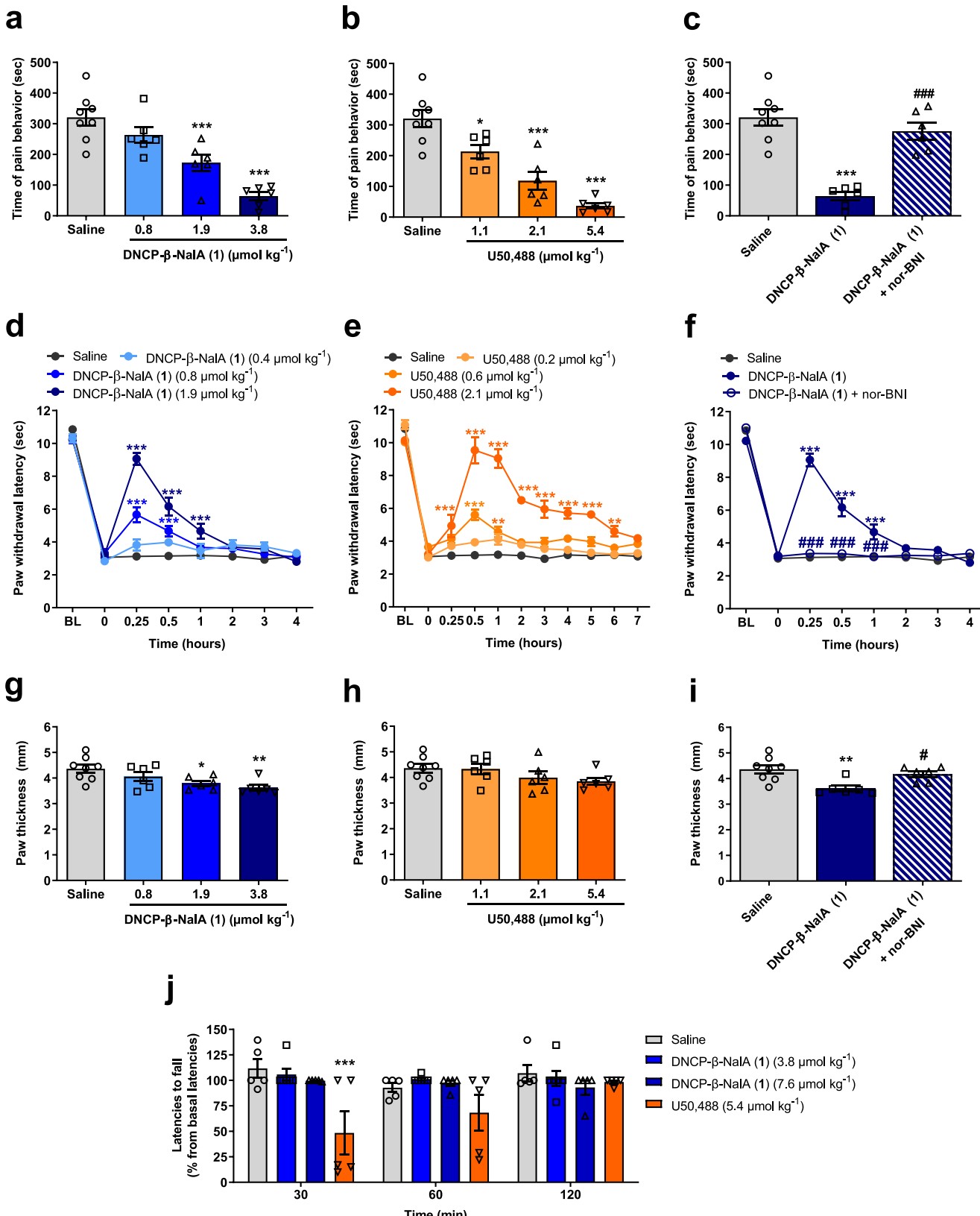

dynamics (MD) simulations, each lasting 500 ns. The simulations' reliability was assessed by analyzing the root mean square deviation of protein backbone atoms (Supplementary Fig. 22) indicating consistent stability throughout. Comparisons of the calculated average distances between Cα-atoms of the receptor and Cα-atoms of the peptide moiety of DNCP-β-NalA(**1**) from the MD simulations and experimentally

derived data from the cryo-EM structure indicate that the peptide portion of the ligand maintained similar interactions pattern to the experimentally determined structure (Supplementary Table 8 and Supplementary Fig. 23). To assess the amino acid residues that are involved in ligand binding throughout the MD simulations, we calculated the fraction of protein amino acid residues within a 4.5 Å distance

**Fig. 4 | In vivo pharmacology of DNCP-β-NalA(1) after s.c. administration in male mice. a, b** Formalin test, dose-dependent effect; groups: saline ($n = 8$ mice), DNCP-β-NalA(**1**) (0.8 μmol kg$^{-1}$, $n = 6$ mice; 1.9 μmol kg$^{-1}$, $n = 6$ mice; 3.8 μmol kg$^{-1}$, $n = 6$ mice) and U50,488 (1.1 μmol kg$^{-1}$, $n = 6$ mice; 2.1 μmol kg$^{-1}$, $n = 6$ mice; 5.4 μmol kg$^{-1}$, $n = 6$ mice); One-way ANOVA, $F_{(3, 22)} = 19.97$, $P < 0.0001$ (**a**), and $F_{(3, 22)} = 27.10$, $P < 0.0001$ (**b**). **c** Formalin test, antagonism by nor-BNI; groups: saline ($n = 8$ mice), DNCP-β-NalA(**1**) (3.8 μmol kg$^{-1}$, $n = 6$ mice) and DNCP-β-NalA(**1**)+nor-BNI (3.8 μmol kg$^{-1}$ + 13.6 μmol kg$^{-1}$, $n = 6$ mice); One-way ANOVA, $F_{(2, 17)} = 29.67$, $P < 0.0001$. **d, e** CFA-induced inflammatory hyperalgesia, dose- and time-dependent effect; groups: saline ($n = 8$ mice), DNCP-β-NalA (**1**) (0.4 μmol kg$^{-1}$, $n = 6$ mice; 0.8 μmol kg$^{-1}$, $n = 6$ mice; 1.9 μmol kg$^{-1}$, $n = 6$ mice) and U50,488 (0.2 μmol kg$^{-1}$, $n = 6$ mice; 0.6 μmol kg$^{-1}$, $n = 7$ mice; 2.1 μmol kg$^{-1}$, $n = 8$ mice); Two-way ANOVA, $F_{(3, 176)} = 46.10$, $P < 0.0001$ (**d**), and $F_{(3, 275)} = 173.0$, $P < 0.0001$ (**e**). **f** CFA-induced inflammatory hyperalgesia, antagonism by nor-BNI; groups: saline ($n = 8$ mice), DNCP-β-NalA(**1**) (1.9 μmol kg$^{-1}$, $n = 6$ mice) and DNCP-β-NalA(**1**)+nor-BNI (1.9 μmol kg$^{-1}$+13.6 μmol kg$^{-1}$, $n = 6$ mice); Two-way ANOVA, $F_{(2, 136)} = 91.61$, $P < 0.0001$. **g, h** Paw thickness, dose-dependent effect; groups: saline ($n = 8$ mice), DNCP-β-NalA(**1**) (0.8 μmol kg$^{-1}$, $n = 6$ mice; 1.9 μmol kg$^{-1}$, $n = 6$ mice; 3.8 μmol kg$^{-1}$, $n = 6$ mice) and U50,488 (1.1 μmol kg$^{-1}$, $n = 6$ mice; 2.1 μmol kg$^{-1}$, $n = 6$ mice; 5.4 μmol kg$^{-1}$, $n = 6$ mice); One-way ANOVA, $F_{(3, 22)} = 5.016$, $P = 0.0084$ (**g**), and $F_{(3, 22)} = 1.770$, $P = 0.1823$. **i** Paw thickness, antagonism by nor-BNI; groups: saline ($n = 8$ mice), DNCP-β-NalA(**1**) (3.8 μmol kg$^{-1}$) and DNCP-β-NalA(**1**)+nor-BNI (3.8 μmol kg$^{-1}$ + 13.6 μmol kg$^{-1}$, $n = 6$ mice); One-way ANOVA, $F_{(2, 17)} = 7.239$, $P = 0.0053$. **j** Rotarod test, motor coordination; groups: saline ($n = 5$ mice), DNCP-β-NalA(**1**) (3.8 μmol kg$^{-1}$, $n = 6$ mice; 7.6 μmol kg$^{-1}$, $n = 5$ mice) and U50,488 (5.4 μmol kg$^{-1}$, $n = 5$ mice); Two-way ANOVA, $F_{(3, 51)} = 7.992$, $P = 0.0002$. One-way ANOVA with Dunnett's (**a, b, g, h**) and Tukey's post hoc test (**c, i**); Two-way ANOVA with Bonferroni's post-hoc test for (**d–f, j**). *$P < 0.05$, **$P < 0.01$, ***$P < 0.001$, drug vs. saline group; #$P < 0.05$, ##$P < 0.001$, DNCP-β-NalA(**1**) vs. DNCP-β-NalA(**1**)+nor-BNI. All data represent means ± s.e.m. Source data are provided as a Source Data file.

from the bound ligand in the combined simulations trajectories (Supplementary Fig. 24). Within binding pocket 1, the ligand demonstrated molecular interactions with Q115$^{2.60}$, L135$^{3.29}$, D138$^{3.32}$, Y139$^{3.33}$, M142$^{3.36}$, V230$^{5.42}$, W287$^{6.48}$, I290$^{6.51}$, I316$^{7.39}$, G319$^{7.42}$, and Y320$^{7.43}$, which align well with the interactions observed in the previously published structure (PDB: 6B73). In peptide-ring binding pocket 2, the ligand primarily engaged in molecular interactions with E209$^{ECL2}$, C210$^{ECL2}$, L212$^{ECL2}$ and Y313$^{7.36}$, as well as amino acid residues S303$^{ECL3}$, H304$^{ECL3}$, A308$^{7.31}$, and L309$^{7.32}$ in strong agreement with the cryo-EM data. The peptide portion of the ligand also had significant contact frequency with amino acid residues located on TM6 and TM7: F293$^{6.54}$, I294$^{6.55}$, E297$^{6.58}$ and Y312$^{7.35}$ (Supplementary Fig. 24).

The aspartic acid residue D138$^{3.32}$ has been known as an anchoring residue important for the binding and signaling of many KOR agonists that usually forms hydrogen-bond (H-bond) interactions with the ligand[37]. It is interesting that D138$^{3.32}$ appears to form a weak salt-bridge interaction (3.8 Å) with DNCP-β-NalA(**1**) and the D138$^{3.32}$N mutation caused an 8-fold loss of potency for DNCP-β-NalA(**1**), but a 1,000-fold for U50,488 (Fig. 5a, Supplementary Figs. 25, 26 and Supplementary Table 9). This is further confirmed by the reduced binding affinity of DNCP-β-NalA(**1**) in KOR D138A or D138N mutants (Supplementary Fig. 27 and Supplementary Table 10). This is in accordance with the MD simulations where the distance between the ligand and D138$^{3.32}$ sampled a wide distance distribution (Supplementary Fig. 28). This could be due to the massive interactions formed between KOR and other parts of DNCP-β-NalA(**1**), as the mutational screening showed that residues from both binding pockets could dramatically affect the potency and/or efficacy of DNCP-β-NalA(**1**) in G protein and β-arrestin activation (Fig. 5b–e, Supplementary Tables 11 and 12). In binding pocket 1 (Fig. 5b), while most of the alanine mutations decreased the agonist activity of DNCP-β-NalA(**1**), the Q115$^{2.60}$A had only a minimal effect on G$_{i1}$ activation, but significantly increased the efficacy of β-arrestin-2 recruitment (60% vs. 35% in WT) (Fig. 5c and Supplementary Table 11). The I135$^{3.29}$A and K227$^{5.39}$A appear to specifically affect DNCP-β-NalA(**1**)'s efficacy in receptor activation (G$_{i1}$ activation, KOR-WT 86%, I135$^{3.29}$A 46%, K227$^{5.39}$A 48%), while not or slightly reduces the potency (Fig. 5c and Supplementary Table 11). The more profound observation was from the mutational analysis of residues in the second binding pocket (Fig. 5d) that may have formed H-bond or hydrophobic interactions (residues within 4 Å of the ligand) with the peptide ring. Several residue mutations, such as E209$^{ECL2}$A, E297$^{6.58}$A and L309$^{7.32}$A led to increased potency in G protein activation and enhanced efficacy in arrestin recruitment (Supplementary Fig. 29, Supplementary Tables 12–14). The role of extracellular vestibule has been investigated in the KOR-dynorphin structure[38], in which mutations of ECL2/3 led to significant reduction of dynorphin A$_{1-13}$'s agonist activity. Comparison of dynorphin with DNCP-β-NalA(**1**) shows overlapping of several binding sites (Supplementary Fig. 29). Characterization of dynorphin in KOR E209$^{ECL2}$A, E297$^{6.58}$A or L309$^{7.32}$A mutants displayed opposite

effects compared to DNCP-β-NalA(**1**), suggesting that the ECL2/3 and extracellular transmembrane ends play important roles in regulating ligand-specific responses during opioid receptor activation (Supplementary Fig. 29, Supplementary Tables 13 and 14). The MD simulations indicate potential interactions between the ligand and amino acid residues (E209$^{ECL2}$, E297$^{6.52}$ and L309$^{7.32}$). In most simulations, the highest probability of distance consistently remained below 6 Å. However, in certain simulations, larger distance probability values were observed, indicating increased flexibility between the ligand and those specific residues (Supplementary Fig. 28). Another residue, Y312$^{7.35}$ (TM7) has been reported to affect KOR's functional selectivity by affecting arrestin signaling[8]. This amino acid residue is involved in hydrophobic contacts with the ligand more than 95% of the simulation trajectory (Supplementary Fig. 24). The Y312$^{7.35}$A mutant activated by DNCP-β-NalA(**1**) was devoid of arrestin recruitment while only a 4-fold loss of potency in G protein activation (Fig. 5e) could be observed. It is important to note that the cell surface expression levels of all mutants were within 2-fold range compared to the wild-type KOR as determined by ELISA (Supplementary Fig. 30).

## Discussion

Our designed thioether cyclized peptides conjugated to β-NalA—a small molecule, partial agonist of the KOR whose structure resembles MP1104[26]—mimic the orthosteric properties of MP1104[8] and simultaneously interact with secondary binding sites on KOR. In contrast to high-throughput screening methods and recent examples of (ultra) large library docking of compounds[39], we overcome multiple rounds of structure-based design and pharmacological testing; we need to synthesize and experimentally characterize only four compounds to discover a high-affinity molecule with patterns of GPCR signaling and pharmacology. This compound, DNCP-β-NalA(**1**), is a full agonist, exhibiting nanomolar KOR potency, with preferential efficacy bias for G proteins over β-arrestins—this is notable as KOR agonists with partial β-arrestin recruitment are often associated with fewer side effects in vivo[31,40,41]. Subtype selectivity on the morphinan template is difficult to achieve (MP1104 is a pan opioid binder)[42]; DNCP-β-NalA(**1**) exhibits ~80-and ~330-fold selectivity for KOR over DOR, and for KOR over NOP receptor, respectively, but is equipotent at MOR. The mixed action of DNCP-β-NalA(**1**) as biased agonist at KOR and competitive antagonist at MOR might thus contribute to the observed antinociceptive efficacy and result in a favorable side effect profile in male mice. The development of mixed KOR agonists/MOR antagonists has been explored as a strategy to develop safer pain medications[12].

The cryo-EM structure of DNCP-β-NalA(**1**) bound to KOR revealed a molecular network of residues involving ECL2/3 and TM6/7 that control the functional selectivity of the receptor. The mutations E209$^{ECL2}$A, E297$^{6.58}$A and L309$^{7.32}$A increase β-arrestin efficacy bias and G$_i$ potency of DNCP-β-NalA(**1**). The TM5-ECL2 region has previously been proposed to modulate bias at other GPCRs such as opioid[29],

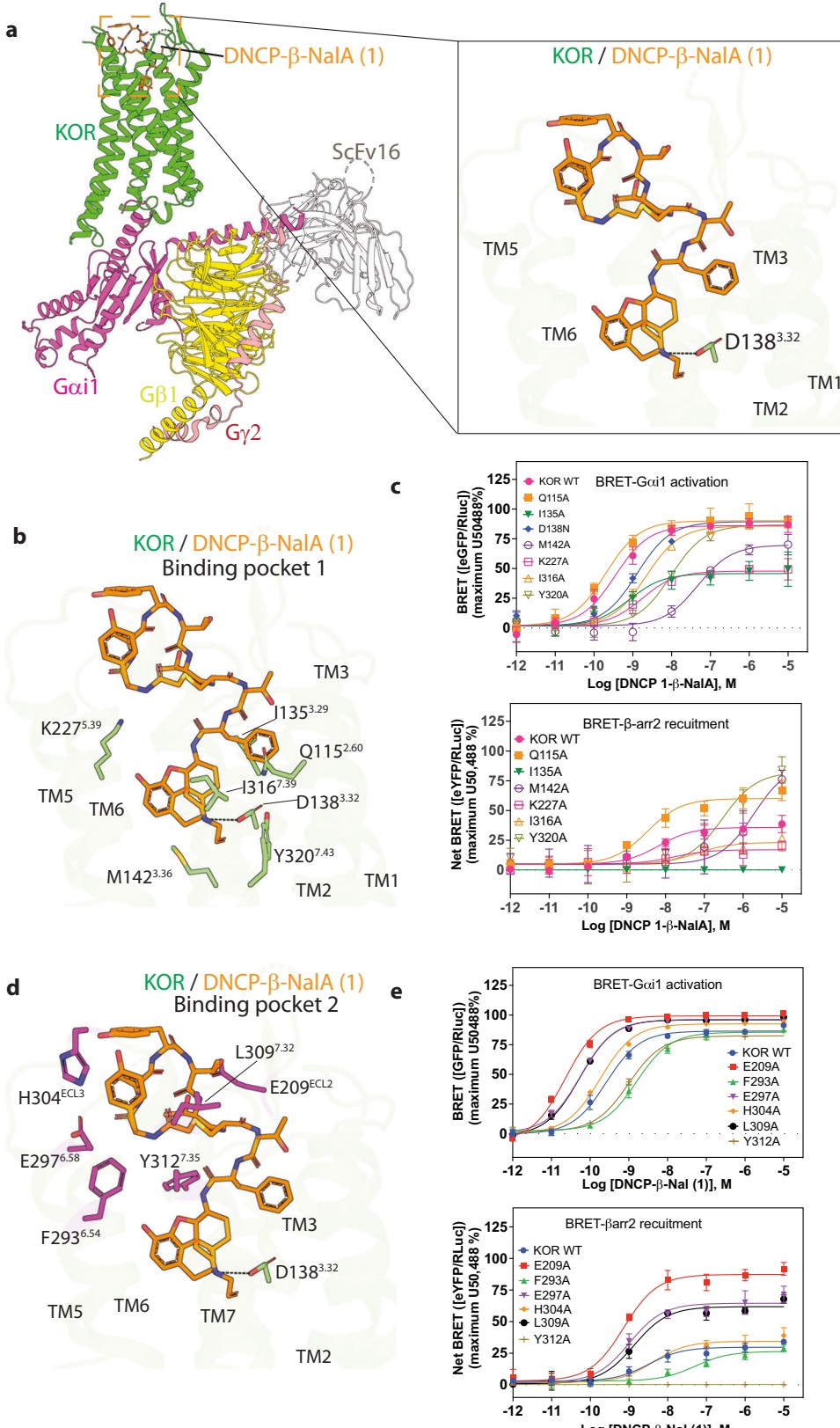

**Fig. 5 | Cryo-EM structure of the KOR-DNCP-β-NalA(1)-G$_{i1}$ complex. a** Overall architecture of the active state human KOR bound to DNCP-β-NalA(**1**) and G-protein heterotrimer (Gα$_{i1}$, Gβ$_1$, Gγ$_2$). The KOR-G-protein complex was further stabilized by a single-chain antibody scFv16. The right panel shows the binding pose of DNCP-β-NalA(**1**) at KOR. The highly conserved anchoring residue D138, as part of the orthosteric binding pocket of KOR is shown. **b** The interactions between the bottom half of DNCP-β-NalA(**1**) and the orthosteric site of KOR. **c** Effects of orthosteric residues on DNCP-β-NalA(**1**)-mediated Gα$_{i1}$ protein and β-arrestin-2 signaling (*n* = 3). For KOR D138N mutant, the reference ligand is salvinorin A because U50,488 is inactive at this mutant. **d** The interactions between the peptide macrocycle of DNCP-β-NalA(**1**) and the extracellular binding pocket 2 of KOR. **e** Effects of binding pocket 2 residues on DNCP-β-NalA(**1**)-mediated Gα$_{i1}$ protein and β-arrestin-2 signaling (*n* = 3). All data are presented as mean values ± s.e.m. Source data are provided as a Source Data file.

serotonin[43] and dopamine[44] receptors. Conversely, Y312[7.35]A, as previously determined for KOR and MOR[29], shifted bias towards the $G_i$ protein and reduced arrestin recruitment. Both U50,488 and MP1104 were previously identified as balanced KOR agonists with ~100% efficacy in the β-arrestin recruitment[8]. While the small molecule portion of DNCP-β-NalA(**1**) adopts a similar conformation to U50,488 and MP1104, the peptide ring forms extensive interactions with ECL2/3 and TM6/7 residues, suggesting that these additional interactions may account for DNCP-β-NalA(**1**)'s G protein-biased or partial agonist activity.

A major challenge in developing functionally specific molecules is the incorporation of multiple receptor conformational states in the design process[45]. The high-resolution cryo-EM structure revealed conformational changes in both the ECL2/3 and the designed peptide. Here, we only utilize the known experimental structure of KOR, which biased the designs toward that conformation; the design protocol can likely be improved by sampling alternative states of the flexible ECLs and peptide using molecular dynamics simulation or other approaches, or by utilizing predicted conformational states of the full receptor. Further improvement of the computational methods and more experimental data should enable design of more selective agonists in the future.

Our design approach coupling a small molecule anchor known to bind KOR, to the free terminus of a lariat-shaped peptide with a cyclic peptide portion making interactions with other portions of the receptor outside the orthosteric site should be broadly applicable to other GPCRs. As illustrated here, the small molecule anchor provides some starting affinity and specificity, which can then be increased and modulated by the interaction of the cyclic peptide portion with the receptor. DNCP-β-NalA(**1**) displays partial β-arrestin recruitment and selectivity for KOR over DOR and NOP receptor, together with KOR-mediated antinociceptive efficacy and anti-inflammatory effects and reduced risk for KOR-related liabilities in male mice. Overall, our approach provides a route to generating diverse sets of high affinity and selective ligands for disentangling the pharmacology of GPCRs and addressing outstanding therapeutic challenges.

## Methods

### Ethical statement

Experiments were performed in male SWISS mice (RjOrl:SWISS; 8–10 weeks old, 30–35 g body weight) purchased from Janvier Labs (Le Genest-Saint-Isle, France). All animal care and experimental procedures were in accordance with the ethical guidelines for the animal welfare standards of the European Communities Council Directive (2010/63/EU) and were approved by the Committee of Animal Care of the Austrian Federal Ministry of Science and Research (Protocol Nr. 2022-0.159.599).

### Computational design

For protein preparation and ligand parametrization the crystal structure of human KOR was obtained from the Protein Data Bank (PDB: 6B73)[8]. KOR was in complex with the epoxymorphinan opioid agonist MP1104 and an active-state-stabilizing nanobody. The nanobody, cholesterol and oleic acid were removed prior to preparation of the structure. To allow MP1104 modeling in Rosetta, we built the necessary ligand parameter file (Supplementary Data 4). MP1104 was modified by removing the iodobenzamide moiety (yielding N-cyclopropylmethyl-epoxy morphinan), leaving behind a free amine (which we refer as CVV, and which corresponds to β-NalA), which enables conjugation to a carboxylic acid. Open Babel[46] was used to generate the necessary input conformers. The mol2 structure of CVV was converted to Rosetta's params file format using the molfile_to_params.py script included with the software. Based on the binding pocket analysis, sampling of hexameric thioether macrocycles was focused on. Backbones were generated through a custom sampling script (Supplementary Data 1)[47,48].

The backbones were clustered using a torsion bin-based clustering method[21]. Prior to docking the cyclic peptide backbones, a two amino acids extension from the free amine of the small molecule anchor (also known as stub) CVV was sampled. The extension was added using PeptideStubMover. All 20 potential amino acids (both D-/L-) were tested as the extension, minimized, and scored. The results were analyzed using python pandas and residues and orientations that had the best interface metrics (high shape complementarity and low ΔΔG) were selected for the extension. The docking was performed using a python script that simply transforms the C,N,O,CA, and CB atoms of the acetylated C-terminal amino acid CYS (known as the 3-letter code: CYY) of each backbone peptide residue onto the corresponding atoms in temporary extended and randomly sampled CYY residue of the anchor (Supplementary Data 2). The docked conformations were then relaxed using Rosetta FastRelax[49] and those with minimal clashes were further carried out for design. Peptide sequences were subsequently designed using Rosetta's FastDesign mover[18]. The details of the design script <kappa_peptide_design.xml> are provided (Supplementary Data 3). The designed peptides were then filtered based on their total Rosetta score, contact molecular surface, shape complementarity, ΔΔG of binding, and contacts at interface using python pandas (Supplementary Fig. 2). The designs in the top 10% of all these metrics were chosen for further analysis. Selected designed models were then visually inspected and those with minimum number of buried unsatisfied hydrogen bond donors and acceptors were selected for synthesis.

### Peptide synthesis

Peptides were synthesized either manually or on an automatic Liberty Blue™ microwave peptide synthesizer (CEM Corporation, NC, USA) (Supplementary Table 1)[50]. DNLPs (**11**–**16**) and DNCPs (**21**–**26**) without small molecule were synthesized as C-terminal amides using Rink amide resin. For DNCPs (**31**–**36**) to be used for conjugation a D-Phe preloaded Wang resin (0.7 mmol g$^{-1}$) was used. After peptide assembly, DNLPs (**11**–**16**) were cleaved from the resin by treatment with Cleavage Cocktail B [trifluoroacetic acid (88% v/v), phenol (5% v/v), water (5% v/v), triisopropylsilane (2% v/v)] for 1 h, followed by diethylether precipitation. To produce the DNCPs, the peptides were subjected cyclized on the resin after assembly using Fmoc-SPPS. First, the N-terminus was bromoacetylated (10 eq. of bromoacetic acid treated with 5 eq. of dicyclohexylcarbodiimide for 1 h), followed by on-resin cleavage of the methoxytrityl group of cysteine (with 2% trifluoroacetic acid in dichloromethane, 12 × 5 min) and finally cyclized to form a thioether (1% N,N-diisopropylethylamine in N,N-dimethylformamide, 18 h). These DNCPs (**21**–**26** and **31**–**36**) were then cleaved from resin by treatment with cleavage cocktail B for 1 h, followed by diethylether precipitation. DNCPs with a C-terminal COOH (**31**–**36**) were coupled to the free $NH_2$ of β-NalA by dissolving the peptide (1 eq.) in dimethylformamide containing HATU (1 eq., 0.4 M) and adding β-NalA (1.1 eq., 25 mM) and diisopropylethylamine (2 eq.) at 0 °C. Reaction mixture was allowed to warm to room temperature and stirred for 30 min. [*Note:* above conc. of stock solutions (0.4 M and 25 mM) were necessary, as at a higher dilution, the conjugation reaction (for DNCP **31**) was incomplete, even after 24 h]. Purification of all peptides was carried out either by RP-HPLC instrument from Waters P150 system with 2545 quaternary gradient module, a Waters 2489 UV/Vis detector and a waters fraction collector III (Waters, MS, USA) or a LC-20AT system equipped with an SPD-20A Prominence UV/VIS detector, and FRC-10A fraction collector using preparative or semipreparative Phenomenex Jupiter $C_{18}$ columns (5 μm, 300 Å, 250 × 21.2 mm or 250 × 10 mm) (Phenomenex, Aschaffenburg, Germany) or a Kromasil $C_{18}$ column (10 μm, 300 Å, 250 × 21.2 mm or 250 × 10 mm) (AkzoNobel, Göteborg, Sweden). Linear gradients from 5% to 65% solvent B (90% acetonitrile, 10% $H_2O$, 0.05% trifluoroacetic acid) and flow rates of 8 mL min$^{-1}$ or 20 mL min$^{-1}$ and 3 mL min$^{-1}$ or 8 mL min$^{-1}$ were applied for preparative

and semipreparative RP-HPLC, respectively. Fractions were collected automatically. Quality of fractions and purified peptides were analyzed by electrospray ionization mass spectrometry (ESI-MS) and analytical RP-HPLC and RP-UPLC (Supplementary Table 2, Supplementary Figs. 4, 5, 7, and 8). For RP-HPLC and RP-UPLC characterization of peptides, a Phenomenex Jupiter $C_{18}$ column (5 µm, 300 Å, 150 × 2 mm) and Phenomenex Luna Omega $C_{18}$ column (1.6 µm 100 Å, 50 × 2.1 mm) were used, respectively. Desired peptide mass was identified by ESI-MS either on LCMS-2020 (Shimadzu, Kyoto, Japan) coupled to an amaZon-SL ion trap or a Brucker Daltonics QqTOF compact (Billerica, MA, USA). For the latter analysis on the QqTOF system a PepMap Acclaim capillary column (Thermo Fisher Scientific; 150 × 0.3 mm, 2 µm, 100 Å) at flow rates of 25 µL min⁻¹ for the loading pump and 6 µL min⁻¹ for the separation pump were used with linear gradients of solvent D (0.08% formic acid in acetonitrile) and solvent C (0.1% formic acid in water).

### Peptide content determination
Peptide concentrations were determined based on peak area detected at 214 nm by analytical RP-HPLC (Phenomenex-Prodigy 100 Å, 5 um, 4.6 × 50 mm, 2 mL min⁻¹, gradient 0–90% B in 10 min) against two peptide standards (oxytocin and vasopressin) with known peptide content established by amino acid analysis. Using the Beer–Lambert law, the peptide concentrations were calculated based on absorbance of standards and samples using calculated extinction coefficients.

### Chemical and reagents
Radioligands [³H]diprenorphine (34.6 Ci mmol⁻¹), [³H]nociceptin (119.4 Ci mmol⁻¹) and guanosine 5′-O-(3-[³⁵S]thio)-triphosphate ([³⁵S] GTPγS, 1,250 Ci mmol⁻¹) were purchased from PerkinElmer (Boston, MA, USA). Guanosine diphosphate (GDP), GTPγS, U50,488, U69,593, DAMGO, DPDPE, DADLE, nor-BNI, nociceptin, CFA, polyethylenimine (PEI), tris(hydroxymethyl) aminomethane (Tris), 2-[4-(2-hydroxyethyl) piperazin-1-yl]ethanesulfonic acid (HEPES), formalin, forskolin, naloxone, bovine serum albumin (BSA), and cell culture media and supplements were obtained from Sigma-Aldrich Chemicals (St. Louis, MO, USA). Opti-MEM was obtained from Gibco-ThermoFisher (Vienna, Austria). Furimazine was obtained from Promega (Madison, USA). Dyn A$_{1-13}$ amide trifluoroacetate salt was purchased from Bachem (Bubendorf, Switzerland). All other chemicals were of analytical grade and obtained from standard commercial sources. DNLPs, DNCPs, DNCP-β-NalA conjugates and β-NalA were solved in water, and further diluted to working concentrations in the appropriate buffer. The cAMP $G_i$ kit was from CisBio-PerkinElmer (Codolet, France) and jetPRIME transfection reagent from Polyplus (Illkirch, France).

### Animals and drug administration
Experiments were performed in male SWISS mice (RjOrl:SWISS; 8–10 weeks old, 30–35 g body weight) purchased from Janvier Labs (Le Genest-Saint-Isle, France). Here, we used male mice to establish pain and behavioral responses as these types of studies are commonly reported using either male or female mice, or both sexes. A total of 124 male SWISS mice were used in the in vivo study. Male mice were group-housed (maximum five animals per cage) in a temperature- (22–23 °C) and humidity-controlled (60–70%) specific pathogen free room with a 12 h light/dark cycle and with free access to food and water. All behavioral experiments were performed during the light cycle. Efforts were made to minimize animal suffering and to reduce the number of mice use. Animals were assigned to groups randomly, and drug treatment experiments were conducted in a blinded fashion. Experimental drugs, DNCP-β-NalA(1), U50,488 and nor-BNI, were prepared in sterile physiological 0.9% saline solution. Test compounds or vehicle (saline) were s.c. administered in a volume of 10 µL g⁻¹ body weight.

### Cell transfection and cloning
Cell transfection was performed with HEK293T (ATCC, CRL-3216) or CHO-K1 (ATCC, CCL-61) cells and jetPRIME transfection reagent according to the manufacturer's protocol. For BRET assay, 1.8 µg of plasmid DNA transiently expressing the mouse KOR tagged fused to EGFP and 0.2 µg of the human β-arrestin-2 coupled to nano luciferase were used. Cloning and generation of stable HEK293T mouse KOR cell line was performed using established molecular and cell biology procedures[51].

### Cell culture and cell membrane preparation
HEK293T mouse KOR, DOR, and MOR stable cell lines were cultured in Dulbecco's Modified Eagle's Medium (DMEM) supplemented with 10% fetal bovine serum (FBS), 50 U mL⁻¹ penicillin/streptomycin and 0.8 mg mL⁻¹ geneticin (G418) and grown at 37 °C and 5% $CO_2$. CHO-K1 cells stably expressing the human MOR or DOR (CHO-hMOR or -hDOR, respectively) were grown at 37 °C in DMEM/Ham's F12 culture medium supplemented with 10% FBS, 0.1% penicillin/streptomycin, 2 mM L-glutamine and 0.4 mg mL⁻¹ geneticin (G418). CHO-hKOR and CHO-hNOP cells were grown at 37 °C in DMEM culture medium supplemented with 10% FBS, 0.1% penicillin/streptomycin, 2 mM L-glutamine and 0.4 mg mL⁻¹ geneticin (G418). All cell cultures were maintained in a humidified atmosphere of 95% air and 5% $CO_2$. Membranes from receptor expressing cells were prepared the following way[52,53]: Exemplarily, HEK-mKOR and CHO-hKOR grown at confluence were removed from the culture plates by scraping, homogenized in 50 mM Tris-HCl buffer (pH 7.7) using a Dounce glass homogenizer, then centrifuged once and washed by an additional centrifugation at 20,000 (HEK-mKOR) or 27,000 × $g$ (CHO-hKOR) for 15 min at 4 °C. Protein content of cell membrane preparations was determined by the method of Bradford using BSA as the standard[54]. All other membranes were prepared accordingly.

### In vitro pharmacological assays
Radioligand binding, cAMP and β-arrestin recruitment assays were performed according to the following protocols[52,55,56]: All reagents and composition of the standard binding buffer were used according to the published protocol[57]. Briefly, displacement radioligand binding assay was performed in a final volume of 300 µL containing [³H] diprenorphine ([³H]DPN), standard binding buffer, competing ligands and membranes from HEK293T cells stably expressing the mouse KOR (7–10 µg/assay), MOR (50–100 µg/assay) or DOR (25 µg/assay). Naloxone (10 µM) was used to determine non-specific binding. Radioligand, naloxone and competing ligands were prepared 4X in the binding buffer. Reaction mixture was incubated at 37 °C for 1 h followed by filtration onto 0.1% polyethlyenimine-soaked GF/C glass fiber filter (Sartorius Stedim, Göttingen, Germany) using a Skatron cell harvester (Skatron AS, Lier, Norway). The radioactivity was measured by liquid scintillation. Radioligand binding assay to the NOP receptor was performed in a final volume of 1 mL in assay buffer (50 mM Tris-HCl buffer pH 7.4 with 1 mg mL⁻¹ BSA) containing [³H]nociceptin (0.1 nM), different concentration of test ligand and membranes from CHO-K1 cells stably expressing the human NOP receptor (15–20 µg/assay). Non-specific binding was determined in the presence of 10 µM nociceptin. Samples were incubated at 25 °C for 1 h followed by filtration onto 0.5% PEI-soaked GF/C glass fiber filter. Filters were washed three times with 5 mL of ice-cold 50 mM Tris-HCl buffer (pH 7.4) using a Brandel M24R cell harvester (Brandel Inc., Gaithersburg, MD, USA). Radioactivity retained on the filters was counted by liquid scintillation counting using a Beckman Coulter LS6500 (Beckman Coulter Inc., Fullerton, CA, USA). The dissociation constant $K_d$ and maximum binding $B_{max}$ values have been previously calculated from the saturation binding studies as 0.87 nM and 7166 fmol mg⁻¹ protein, respectively. The $K_d$ values for MOR (0.81 nM), DOR (1.7 nM) and NOP receptor (0.17 nM) have been used from previously published literature[8,52,58].

The [$^{35}$S]GTPγS binging assay was performed with membranes from CHO-K1 stably expressing the human KOR, DOR or MOR according to the published procedure[53]. Cell membranes (15 μg) in 20 mM HEPES buffer (pH 7.4) supplemented with 10 mM MgCl$_2$ and 100 mM NaCl were incubated with 0.05 nM [$^{35}$S]GTPγS, 10 μM GDP and various concentrations of test compounds in a final volume of 1 mL for 60 min at 25 °C. Antagonism of DNCP-β-NalA(**1**) to the MOR was evaluated by Schild regression analysis, with concentration-response curves of DAMGO measured in the absence and presence of DNCP-β-NalA(**1**) (10, 30 and 100 μM). Non-specific binding was determined using 10 μM GTPγS, and the basal binding was determined in the absence of the test ligand. Samples were filtered over Whatman GF/B glass fiber filters. Filters were washed three times with 5 mL of ice-cold 50 mM Tris-HCl buffer (pH 7.4) using a Brandel M24R cell harvester (Brandel Inc., Gaithersburg, MD, USA). Radioactivity retained on the filters was counted by liquid scintillation counting using a Beckman Coulter LS6500 (Beckman Coulter Inc., Fullerton, CA, USA).

Functional cAMP assay was performed in a final volume of 20 μL using HEK293T cells stably expressing the mouse KOR[58]. A number of 2000 cells were seeded into white 384-well plate and incubated with different concentrations of peptide solutions prepared (2X) in 1X stimulation buffer and forskolin (10 μM). The reaction mixture was incubated at 37 °C for 30 min and the reaction was terminated by adding cryptate-labeled cAMP and cAMP d2-labeled antibody. Following an incubation for 1 h at room temperature, cellular cAMP levels were quantified by homogenous time-resolved fluorescence resonance energy transfer (HTRF, ratio 665/620 nm) on a Flexstation 3 (Molecular Devices, San Jose, USA). Bioluminescence resonance energy transfer (BRET)-based β-arrestin-1/2 recruitment was conducted in a final volume of 200 μL using HEK293 cells transiently co-expressing the human β-arrestin-1/2-nano luciferase and the mouse KOR-EGFP (1:10 ratio). After 16–24 h post-transfection, 50,000 cells per well in phenol red-free DMEM supplemented with 10% FBS were transferred into white clear bottom plates and incubated overnight at 37 °C. The following day, cells were subject to serum starvation for 1 h at 37 °C in phenol red-free DMEM. Furimazine, diluted 1:50 in Hank's balanced salt solution (HBSS) was used as a substrate for nano luciferase. Peptide solutions with indicating concentrations were prepared 4X in HBSS. Prior to establishing the baseline by measuring the signal following addition of furimazine for 5 min at 37 °C, cells were incubated with furimazine for 5 min at 37 °C to allow an efficient cell entry. Subsequently, peptides were incubated at 37 °C for 5 min and the signal was measured on a Flexstation 3 (Molecular Devices, San Jose, USA) for both luminescence at 460 nm (nano luciferase) and fluorescence at 510 nm (EGFP).

To measure KOR-mediated G protein association of Gα$_{i1}$-, Gα$_{i2}$-, Gα$_{i3}$-, Gα$_{oA}$-, Gα$_{oB}$-, Gα$_z$- and Gα$_{gustducin}$ containing heterotrimeric G proteins, HEK293T (ATCC, CRL-3216) cells were co-transfected in a 1:1:1:1 ratio with the mouse KOR and the optimal Gα-RLuc8, Gβ, and Gγ-GFP2 subunits according to ref. 35. Briefly, TransIT-2020 (Mirus Bio LLC, Madison, WI, USA) was used to complex the DNA at a ratio of 3 μL Transit per μg DNA, in Opti-MEM at a concentration of 10 ng DNA per μL Opti-MEM. After 16 h, transfected cells were plated in poly-lysine coated 96-well white clear bottom cell culture plates in plating media (DMEM + 1% dialyzed FBS) at a density of 40–50,000 cells in 100 μL per well and incubated overnight. The next day, media was vacuum aspirated, and cells washed twice with 60 μL of assay buffer (20 mM HEPES, 1X HBSS, pH 7.4). Next, 60 μL of the RLuc substrate, coelenterazine 400a (Nanolight Technologies, 5 μM final concentration in assay buffer) was added per well and incubated for 5 min to allow for substrate diffusion. Afterward, 30 μL of test compound (3X) in buffer (20 mM HEPES, 1X HBSS, 0.3% BSA, pH 7.4) were added per well and incubated for another 5 min. Plates were immediately read for both luminescence at 395 nm and fluorescent GFP2 emission at 510 nm for 1 s per well using the FlexStation plate reader. BRET ratios were calculated as the ratio of the GFP2 emission to RLuc8 emission, where the vehicle-treated cell sample represents the background, eliminating the requirement for measuring a donor-only control sample.

## Serum stability assay

Blood was drawn from healthy volunteers devoid of any medication for 14 days from the antecubital vein using 21 G needles and collected in a serum clot activator tube (4 mL tubes, Greiner bio-one). The study was approved by the Ethical Board of the Medical University of Vienna (EK1548/2020) and conformed to institutional guidelines as well as the Declaration of Helsinki. Volunteers gave written informed consent before blood donation. No sex or gender analysis was conducted as human active serum is expected to have equal activity between the sexes. Blood samples were collected into Vacuette® clot activator tubes (Greiner Bio-One, Austria). After clotting of about 2 h, the samples were centrifuged at 2200 × g for 15 min at 22 °C. The serum was removed and centrifuged again to remove lipids and cell particles at 16,000 × g at 4 °C for 20 min. Peptides, DNCP-β-NalA(**1**) and dyn A$_{1-13}$ (Bachem, Switzerland), were prepared as 500 μM stocks in PBS buffer and diluted 1:10 into the serum sample to give a final concentration of 50 μM. The samples were incubated at 37 °C under continuous shaking at 300 rotations-per-minute. At time points zero, 5, 10, 20, 40, 60, 120, 240, 360, 1440 and 2880 min aliquots of 30 μL were transferred into a low protein binding reaction tube containing protein solubilization buffer (6 M urea in water). The samples were mixed well and incubated on ice for 10 min. For the protein precipitation, 30 μL of a 20% (w/v) trichloroacetic acid was added and incubated on ice for 15 min. The clear supernatant was harvested by centrifugation at 16,000 × g at 4 °C for 20 min and this sample was used for HPLC analysis. The chromatography HPLC system was a ThermoFisher Ultimate 3000 equipped with a pump, autosampler and single wavelength detector. A Phenomenex Kinetex C$_{18}$ 150 × 3 mm, 2.6 μm 110 Å was used as separation column with mobile phases 0.1% trifluoroacetic acid in water and acetonitrile/water trifluoroacetic acid 90/10/0.1% (v/v/v) with a linear separation gradient over 30 min. For the evaluation of the remaining peptide peak over time of incubation in active human serum, the area under the curve (AUC) was used with a detection at 214 nm. The remaining peptide was calculated relative to the AUC at time point zero. The data points were fitted to a one-phase decay function and to determine the resulting half-life of the peptide in serum.

## Generation of constructs for cryo-EM

The human KOR used a construct same as previously determined active-state KOR[8]. Briefly, the construct (a) lacks N-terminal residues 1–53, (b) lacks C-terminal residues 359–380, (c) contains M1-L106 of the thermostabilized apocytochrome b562 RIL (BRIL) from *E. coli* in place of receptor N-terminus residues M1-H53. This N-terminal Bril will be removed using a PreScission cleavage site in the end. The single chain Fab scFv16 has the same sequence as previously reported[59]. A 6xHis tag was added to the C-terminal scFv16 sequence with a PreScission cleavage site inserted between. For the Gα$_{i1}$ protein heterotrimer, Gα$_{i1}$ protein construct was engineered[60] for the binding of scFv16, and then subcloned into a designed vector that co-expresses the Gβ$_1$ and Gγ$_2$[61]. Further modifications were made to enable a stable complex between KOR, Gα$_{i1}$ protein heterotrimer, and scFv16. Specifically, Gα$_{i1}$-dominant negative variants are S47N, E245A, G203A, and A326S.

## Expression of KOR-Gα$_{i1}$ protein-scFv16 complex

The Bac-to-Bac Baculovirus Expression System was applied to generate high-quality recombinant baculovirus for protein expression[8]. For the expression of KOR-Gα$_{i1}$-scFv16 protein complex, Gα$_{i1}$, Gβ$_1$, and Gγ$_2$ were coexpressed with KOR and scFv16, respectively, by infection *Spodoptera frugiperda* Sf9 cells (Expression Systems, 94-001 S) at a cell density of 2.5 × 10$^6$ cells per mL in ESF921 medium (Expression

Systems, Davies, CA, USA) with the P1 baculovirus at a multiplicity of infection (MOI) ratio of 2:2:0.5. Cells were harvested by centrifugation after infection for 48 h (125 rotations-per-minute at 27 °C), washed with HN buffer (25 mM HEPES pH 7.4, 100 mM NaCl), and stored at −80 °C for further purification.

## Purification of the KOR-G$_{i1}$ protein-scFv16 complex
The thawed cell pellet was incubated in buffer containing 20 mM HEPES 7.5, 50 mM NaCl, 1 mM MgCl$_2$, 2.5 U apyrase (NEB), 10 μM agonist (final concentration) and protease inhibitors (500 mM AEBSF, 1 mM E-64, 1 mM leupeptin, 150 nM aprotinin) for 1.5 h at room temperature. Then, harvested the membrane by centrifugation at 64,300 × g for 30 min at 4 °C. The membrane was solubilized in the buffer (40 mM HEPES pH7.5, 100 mM NaCl, 5% (w/v) glycerol, 0.6% (w/v) lauryl maltose neopentyl glycol (LMNG), 0.06% (w/v) cholesteryl hemisuccinate (CHS), 10 μM agonist and protease inhibitors) with 200 μg scFv16 in the cold room. After 5 h, the supernatant was collected by centrifugation at 92,600 × g for 30 min at 4 °C and incubated with 1 mL TALON IMAC resin (Clontech, now Takara Bio Inc, San Jose, CA, USA) and 20 mM imidazole overnight in the cold room. The next day, the resin was collected and washed with 10 ml buffer containing 20 mM HEPES pH 7.5, 100 mM NaCl, 30 mM imidazole, 0.01% (w/v) LMNG, 0.001% (w/v) CHS, 5% glycerol and 5 μM agonist. Then the protein was eluted with the same buffer supplemented with 300 mM imidazole, concentrated and further purified by size-exclusion chromatography (SEC) on the Superdex 200 increase 10/300 column (GE healthcare, WI, USA) that is pre-equilibrated with 20 mM HEPES pH7.5, 100 mM NaCl, 100 μM TCEP, 0.00075% (w/v) LMNG, 0.00025 (w/v) glyco-diosgenin (GDN) and 0.00075% (w/v) CHS, 1 μM agonist. Peak fractions were collected, concentrated and incubated with PNGase F (NEB, MA, USA), PreScission protease (GeneScript, NJ, USA) to remove the potential glycosylation and N-terminal his-BRIL, respectively, and 100 μg scFv16 at 4 °C overnight. In the next day, cleaved his-Bril and protein, uncleaved protein and proteases were separated by the same procedure as described above. Peak fractions were concentrated to 3–5 mg mL$^{-1}$ for electron microscope studies.

## Expression and purification of scFv16 protein
The scFv16 protein was expressed by infection Sf9 cells (Expression Systems 94-001 S). at a cell density of 2.5 × 10$^6$ cells per mL in ESF921 medium (Expression Systems) with the P1 baculovirus at a MOI of 2. After 96 h, the cell cultured medium containing secreted scFv16 protein was collected by centrifugation at 3739 × g for 15 min. The pH of supernatant was adjusted to 7.5 by addition of Tris-base power. Chelating agents were quenched by addition of 1 mM nickel chloride and 5 mM calcium chloride and incubation with stirring for 1 h at room temperature and 5 h in the cold room further. Removed the precipitates by centrifugation and the resultant supernatant was further cleaned with 0.45 μm filter paper and incubated with 2 mL Ni-NTA resin and 10 mM imidazole overnight in the cold room. The Ni-NTA resin was wash next day with 20 mL buffer of 20 mM HEPES pH7.5, 100 mM NaCl, 0.00075% (w/v) LMNG, 0.000075% (w/v) CHS, 0.00025% (w/v) GDN, 20 mM imidazole. The protein was eluted with the same buffer supplemented with 300 mM imidazole, concentrated and further purified on a Superdex 200 increase 10/300 column. Monomeric fractions were pooled, concentrated, flash frozen in liquid nitrogen and stored at −80 °C until future use.

## Cryo-EM data collection and 3D reconstruction
The purified samples (3–4 μL) were applied to glow-discharged 300-mesh Au grids (Quantifoil R1.2/1.3) individually and vitrified using a Vitt mark IV (ThermoFisher). Cryo-EM imaging was performed on a Talos Artica operated at 200 kV at a nominal magnification of 45,000-times using a Gatan K3 direct electron detector at a physical pixel size of 0.88 Å. Each stack movie was recorded for 2 to 2.7 s in 60 frames at a

dose rate of -15 e$^−$/pix/sec. Movies were collected automatically with SerialEM63 using an optimized multishot array procedure[62]. Dose-fractioned image stacks were subjected to the beam-induced motion correction followed by contrast transfer function (CTF) estimation. Particles were selected by Blob particle picker, extracted from micrograph and then used for 2D classification, 3D classification followed by non-uniform refinement. All these steps were performed in cryoSPARC[63,64].

## Model building and refinement
Maps from cryoSPARC were used for map building, refinement and subsequent structural interpretation. The dominant-negative Gα$_{i1}$ trimer model and scFv16 model were adapted from the cryo-EM structure of the MRGPRX2−G Gα$_{i1}$ complex (PDB: 7S8M)[65]. The receptor KOR model was taken from the active-state KOR-Nb39 structure (PDB: 6B73)[8]. The receptor, Gα$_{i1}$ protein, and scFv16 were docked into the cryo-EM map using Chimera[66]. The complex model (KOR-Gα$_{i1}$ protein-scFv16) were manually built in Coot[67], followed by several rounds of real-space refinement using Phenix[68]. The model statistics were validated using Molprobity[69]. Structural figures were prepared using Chimera or Pymol (https://pymol.org/2/).

## Molecular dynamics simulations
We performed an all-atomistic molecular dynamics simulation of KOR-DNCP-β-NalA(1) complex using the Compute Unified Device Architecture (CUDA) version of particle-mesh Ewald molecular dynamics (PMEMD) on graphics processing units (GPUs) in AMBER22[70]. The initial coordinate of G proteins (Gα$_{i1}$/Gβ$_1$/Gγ$_2$) and KOR bound with DNCP-β-NalA(1) is reported in this manuscript. The ICL and ECL missing loops of KOR are modeled by aligning KOR on PDB: 8F7W[38] as template structure. The missing domain of guanine nucleotide-binding protein Gα$_{i1}$ is modeled using Alpha Fold (UniProt: P63096)[71]. All missing atoms in Gβ$_1$, Gγ$_2$ proteins and KOR are added using TLEAP implemented in AMBER22 and the missing hydrogens in DNCP-β-NalA(1) are added by Open-Babel[46]. Then the model G proteins-KOR-DNCP-β-NalA(1) complex was embedded in a lipid bilayer membrane using PACKMOL-Memgen module[72] of AmberTool23[70]. The lipid composition 0.55:0.15:0.30 ratio is used for the construction of lipid bilayer membrane in the ratio of 1,2-dipalmitoyl-sn-glycero-3-phosphatidylcholine (DPPC), 1,2-dioleoyl-sn-glycero-3-phosphatidylcholine (DOPC) and cholesterol (CHL1), respectively. Similar lipid composition ratio was reported previously[73]. The final KOR-DNCP-β-NalA(1) complex with G proteins contained 330 DPPC, 90 DOPC and 180 CHL1 lipids, 63,537 water molecules, and 172 sodium and 172 chloride ions (Supplementary Table 15). The Amberff19SB[74] and Lipid21[75] force field parameters are used for protein and lipid molecules, respectively. The DNCP-β-NalA(1) interacting potential parameters are generated using GAFF2 force field[76] and the columbic charge of DNCP-β-NalA(1) atoms is assigned through Antechamber at AM1BCC scheme. Water is modeled using TIP3P water model[77] and counter-ions are model using the Joung/Cheatham parameters[78]. The system is minimized for 10,000 steps, the first 5000 step using steepest descent method and the next 5000 steps using conjugate gradient method using 5 kcal$^{-1}$ mol$^{-1}$ Å$^{-2}$ on non-solvent molecules. The minimized structure is heated from a 0 K to 310 K in 100 ps using NVT ensemble by putting restraints of 5 kcal mol$^{-1}$ Å$^{-2}$ on non-solvent molecules, followed by 1 ns NPT equilibration with restraints on non-solvent molecules. The system was further equilibrated for another 30 ns a in NPT ensemble using 5 kcal mol$^{-1}$ Å$^{-2}$ restraint only on backbone of G proteins and KOR-DNCP-β-NalA(1) complex allowing equilibration of lipid and solvent molecules. The restraints were removed in 5 steps from 5 kcal mol$^{-1}$ Å$^{-2}$ to 0 kcal mol$^{-1}$ Å$^{-2}$ in a stepwise of 2 ns each step. The 100 ns equilibration run is performed without restraints at 1 atm pressure and 310 K temperature, using the Langevin thermostat with collision frequency of 2 ps$^{-1}$ and the Monte Carlo barostat with semi-isotropic

pressure scaling using 2 fs time step. The particle-mesh Ewald (PME) method is used to calculate long-range electrostatic interactions. A cut-off 10 Å is used for all non-bonded interactions. Four independent 500 ns production run is started from last frame of equilibrated system using identical setting except the hydrogen mass repartitioning was used at a 4 fs time step[79]. The trajectories were saved at an interval of 10 ps. AmberTools 23[70] and VMD[80] were used for analysis and visualization, respectively. The fraction of contacts between DNCP-β-NalA(**1**) and the KOR is used to quantify inter-atomic interactions within the complex. These contacts are defined by applying a distance cutoff of 4.5 Å between all heavy atoms of DNCP-β-NalA(**1**) and all heavy atoms of amino acids in KOR. Hydrogen bonds between residue pairs are calculated using a criteria of acceptor-to-donor heavy atom distance of 3.5 Å and an angle of 135°. The combined 2 μs simulation trajectories are clustered together using the k-means clustering algorithm based on DNCP-β-NalA(**1**) heavy atom root-mean-square deviation (RMSD) to extract a representative conformation of the KOR-DNCP-β-NalA(**1**) complex, implemented in AmberTools23.

### Enzyme-linked immunosorbent assay (ELISA)

To determine the expression level of constructs KOR and its mutants, HEK293T cells were plated in poly-L-lysine-coated 96-well white clear bottom cell culture plate at a density of 20,000–30,000 cells per well overnight and transfected with 300 ng plasmids for each well. After 24 h, plates were decanted and fixed with 4% (w/v) paraformaldehyde for 10 min at room temperature. Then, cells were washed twice with 1X PBS (phosphate buffered saline, pH 7.4) and blocked by 1X PBS containing 0.5% (w/v) non-fat milk for at least 30 min at room temperature followed by incubation with anti-Flag (M2)−horseradish peroxidase-conjugated antibody (Sigma-Aldrich, A8592) diluted 1:20,000 in the same buffer for 1 h at room temperature. After washing three times with 1X PBS, 1-StepTM Ultra-TMB ELISA substrate (ThermoFisher, 34028) was added to the plates and incubated at 37 °C for 15–30 min and terminated by addition of 1 M sulfuric acid ($H_2SO_4$) stop solution. Finally, plates were read at the wavelength of 450 nm with the BioTek Luminescence reader.

### In vivo pharmacology

The formalin test[81] was performed in male SWISS mice (8–10 weeks old, 30–35 g body weight, $n = 50$ mice). Following a habituation period of 15 min to individual transparent observation chambers, male mice were s.c. administered saline (control), DNCP-β-NalA(**1**) (0.8, 1.9 and 3.8 μmol kg⁻¹) or U50,488 (1.1, 2.1 and 5.4 μmol kg⁻¹), 15 min prior injection of 20 μL of 5% formalin aqueous solution to the plantar surface of the right hindpaw. The time (in sec) each animal spent licking, biting, lifting, and flinching of the formalin-injected paw (pain behavior) was recorded in 5 min intervals between 15 and 30 min after the injection of formalin (Phase II reaction). For the antagonism study, nor-BNI (13.6 μmol kg⁻¹) was s.c. administered 24 h before DNCP-β-NalA(**1**) (3.8 μmol kg⁻¹) and pain behavior was assessed as described above. Antinociceptive activity in the formalin test, as percentage decrease in duration of pain behavior compared to the saline (control) group, was calculated according to the following formula: $100*[\frac{C-T}{C}]$, where C is the mean time in control (saline) group and T is the time in drug-treated group[82]. Dose-response relationships of percentage inhibition of pain behavior were constructed, and the dose necessary to produce a 50% effect ($ED_{50}$) and 95% confidence limits (95% CL) was calculated according to the method of Litchfield and Wilcoxon[83].

Sixty min after injection of formalin to male mice, maximal paw thickness of the right hind paw (i.e., formalin-injected hind paw) was measured using a digital micrometer (FixPoint, Vienna, Austria). Data are expressed in mm.

CFA-induced inflammatory hyperalgesia was induced in male SWISS mice (8–10 weeks old, 30–35 g body weight, $n = 53$ mice) by s.c.

injection of 20 μL emulsified CFA (1 mg mL⁻¹) into the plantar surface of the right hindpaw under brief isoflurane anesthesia[52]. Nociceptive testing was performed 72 h after inoculation with CFA. Hindpaws withdrawal latencies to thermal stimulation were measured using the Hargreaves test[84] with an analgesiometer (Ugo Basile S.R.L., Varese, Italy). Male mice were habituated in individual Plexiglas boxes positioned on a glass surface (Ugo Basile S.R.L., Varese, Italy) for three days prior to the inoculation with CFA. A movable infrared generator (30% intensity) located under the glass floor was focused onto the plantar surface of the hindpaw and switched on to heat. Onset of the radiant stimulus triggered a timer, which was stopped by subsequent paw movement. To prevent tissue damage, a cut-off time of 20 s was imposed. Thermal sensitivity was measured before inoculation with CFA into the right hindpaw (basal latencies, BL), 72 h post-inoculation (pre-treatment values, defined as 0 h), and at different time points after s.c. administration of saline (control), DNCP-β-NalA(**1**) (0.4, 0.8 and 1.9 μmol kg⁻¹) or U50,488 (0.2, 0.6 and 2.1 μmol kg⁻¹). For the antagonism study, nor-BNI (13.6 μmol kg⁻¹) was s.c. administered 24 h before DNCP-β-NalA(**1**) (1.9 μmol kg⁻¹) and pain behavior was assessed. The paw withdrawal latencies to thermal stimulus was expressed in seconds.

The rotarod test was used to evaluate coordinated locomotion and potential sedation induced by test drugs[85]. The accelerating rotarod treadmill (Acceler Rota-Rod 7650, Ugo Basile s.r.l., Varese, Italy) for mice (diameter 3.5 cm) was used. Male SWISS mice (8–10 weeks old, 30–35 g body weight, $n = 21$ mice) were habituated to the equipment in two training sessions (30 min apart) one day before testing. On the experimental day, male mice were placed on the rotarod, and treadmill was accelerated from 4 to 40 rotations-per-minute over a period of 5 min. The time spent on the drum was recorded for each mouse before (baseline) and at 30, 60 and 120 min after s.c. administration of saline (control), DNCP-β-NalA(**1**) (3.8 and 7.6 μmol kg⁻¹) or U50,488 (5.4 μmol kg⁻¹). Decreased latencies to fall in the rotarod test indicate impaired motor performance. A 300 s cut-off time was used. The rotarod data are expressed as percentage (%) changes from the rotarod latencies obtained before (baseline) and after drug administration (test, T) were calculated as $100*(\frac{T}{B})$[53].

### Data analysis

In vitro and in vivo data analyses were carried out with GraphPad Prism (GraphPad Software, San Diego). Data are presented as means ± s.e.m (unless otherwise stated). Data from cAMP or β-arrestin recruitment or G protein biosensor assays were fitted to three-parameter non-linear regression curves with a bottom constrained to zero, a slope of one and logarithmic scale. The observed sigmoidal curves were normalized to 100% which refers to the saturating concentration of full KOR agonists U50,488, U69,593 or dyn $A_{1-13}$ (for KOR), DAMGO (for MOR) and DPDPE or DADLE (for DOR) (10 μM each) (unless otherwise stated). Data from radioligand binding studies were fitted to a three-parameter logistic Hill equation to derive the $IC_{50}$ and inhibition constants ($K_i$) calculated by the approximation of Cheng and Prusoff[86]. Specific binding data were normalized to 100% of [³H]DPN (KOR, MOR, and DOR) or [³H]nociceptin (NOP receptor) (no competing ligand). The 100% defines an average of 4000–6000 fmol mg⁻¹ of protein per assay for KOR, 500–1000 fmol mg⁻¹ of protein for MOR, 1500 fmol mg⁻¹ of protein for DOR and 150–300 fmol mg⁻¹ for NOP receptor. The increase in [³⁵S]GTPγS binding above the basal activity was used to determine potency ($EC_{50}$, in nM) and efficacy (as $E_{max}$ in %, as maximum stimulation relative to the reference full KOR agonist U69,593) from concentration-response curves by nonlinear regression analysis. For Schild regression analysis, data were normalized to 100%, i.e., the highest concentration (10 μM) of DAMGO and fitted to three-parameter non-linear regression curves. The logarithm of the concentration-ratio $\frac{A'}{A-1}$ (A denotes $EC_{50}$ of DAMGO in presence of DNCP-β-NalA(**1**); A′ is $EC_{50}$ of DAMGO) was plotted against the

logarithm of the respective concentration of antagonist DNCP-β-NalA(**1**) to derive the pA2 value (functional inhibitory affinity). All in vitro assays were performed in duplicates or triplicates and repeated at least three times with independently prepared samples. In vivo data were statistically evaluated using one-way ANOVA with Dunnett's or Tukey's multiple comparison post hoc test, or two-way ANOVA with Bonferroni's post hoc test with significance set at $P < 0.05$.

## Reporting summary

Further information on research design is available in the Nature Portfolio Reporting Summary linked to this article.

## Data availability

The data supporting this study are available from the corresponding authors upon request. The cryo-EM maps and corresponding coordinates have been deposited in the Electron Microscopy Data Bank (EMDB) under accession codes EMD-29026 and the Protein Data Bank (PDB) under accession codes 8FEG. The script data generated in this study are provided as Supplementary Data files. Source data are provided with this paper in the Source Data file and from the corresponding authors upon request. Source data are provided with this paper.

## Code availability

The codes used for design and instructions on how to run, as well as the Molecular Dynamics (MD) simulation initial coordinate, simulation input files and a coordinate file of the final output can be found in kappa_SMpeptide_design folder in a github repository (https://github.com/kdeibler/kappa_SMpeptide_deisgn; generated https://doi.org/10.5281/zenodo.10070208).

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

## Acknowledgements

We would like to thank Bryan Roth (UNC, Chapel Hill, NC) for sharing the human KOR plasmid. HEK293T cells stably expressing the mouse MOR and DOR were a kind gift of Dr. Oliver Kudlacek (Medical University of Vienna, Austria). CHO-K1 cells stably expressing the human opioid receptors were kindly provided by Dr. Lawrence Toll (SRI International, Menlo Park, CA). Research of C.W.G. (P32109), M.S. (I4697), R.H. and K.B.J. (ZK-81B) are supported by the Austrian Science Fund (FWF). C.W.G. has been supported by the Austrian Federal Ministry for Labor and Economy and the Federal Ministry for Climate Action, Environment, Energy, Mobility, Innovation and Technology with an AWS Prize project (P2387582). E.M. was a Marietta Blau Fellow of Austrian Federal Ministry of Education, Science and Research (ICM-2019-13441). K.D. and T.C. were Washington Research Foundation Fellows. Research of M.M. was supported by an ERC start grant (FA 705007). Research in the laboratory of D.J.C. is supported by the National Health and Medical Research Council Australia (2009564) and the Australian Research Council (CE200100012).

## Author contributions

C.W.G. and G.B. initiated the project. K.D. performed computational peptide design. T.W.C. generated initial parameter files. E.M., R.H., J.K., and K.B.J. performed peptide synthesis and analysis. E.M., N.T., J.H., K.A., A.-L.O.-M., and N.H. conducted in vitro pharmacological studies. M.H.R. and L.H. performed and analyzed the MD simulations. A.-L.O.-M. and M.S. conducted in vivo pharmacological studies. J.H., J.F.F., and T.C. performed protein purification, cryo-EM studies and structural analysis. J.K., B.R.V., D.J.C., M.M., S.M., M.S., P.H., and D.B. contributed reagents and tools. All authors analyzed data. All authors contributed to drafting the manuscript. E.M., K.D., T.C., D.B., and C.W.G. edited the manuscript for submission.

## Competing interests

S.M. is a co-founder of Sparian Biosciences. The other authors declare no competing interests.

## Additional information

¹Center for Physiology and Pharmacology, Institute of Pharmacology, Medical University of Vienna, 1090 Vienna, Austria. ²Institute for Protein Design, University of Washington, Seattle, WA 98195, USA. ³Center for Clinical Pharmacology, University of Health Sciences & Pharmacy at St. Louis and Washington University School of Medicine, St. Louis, MO 63110, USA. ⁴Institute of Biological Chemistry, Faculty of Chemistry, University of Vienna, 1090 Vienna, Austria. ⁵Department of Pharmaceutical Chemistry, Institute of Pharmacy and Center for Molecular Biosciences Innsbruck (CMBI), University of Innsbruck, Innrain 80-82, 6020 Innsbruck, Austria. ⁶Institute for Molecular Bioscience, Australian Research Council Centre of Excellence for Innovations in Peptide and Protein Science, The University of Queensland, Brisbane, QLD 4072, Australia. ⁷Department of Biochemistry and Molecular Biology, University of Maryland Baltimore, Baltimore, MD 21201, USA. ⁸Department of Pharmaceutical and Administrative Sciences, Saint Louis College of Pharmacy, University of Health Sciences & Pharmacy in St. Louis, St. Louis, MO 63110, USA. ⁹Department of Anesthesiology, Washington University School of Medicine, St. Louis, MO 63110, USA. ¹⁰Institute for Molecular Bioscience, The University of Queensland, Brisbane, QLD 4072, Australia. ¹¹Department of Bioengineering, Knight Campus, University of Oregon, Eugene, OR 97403, USA. ¹²Department of Biochemistry, University of Washington, Seattle, WA 98195, USA. ¹³Howard Hughes Medical Institute, University of Washington, Seattle, Washington, WA 98195, USA. ¹⁴Present address: Institute for Protein Design, University of Washington, Seattle, WA 98195, USA. ¹⁵Present address: Novo Nordisk Research Center Seattle, Novo Nordisk A/S, 530 Fairview Ave N #5000, Seattle, WA 97403, USA. ¹⁶Present address: School of Biomedical Sciences, Faculty for Medicine, The University of Queensland, Brisbane, QLD 4072, Australia. ¹⁷These authors contributed equally: Edin Muratspahić, Kristine Deibler, Jianming Han. ✉e-mail: taoche@wustl.edu; dabaker@uw.edu; christian.w.gruber@meduniwien.ac.at

