## [Peer Review File · Nature Communications]

Reviewers' Comments:

Reviewer #1:

Remarks to the Author:

In the study, the authors present new peptide-small molecule conjugates that act as kappa-opioid receptor (KOR) ligands. Recent studies suggested, that the KOR is an attractive therapeutic target for non-addictive pain killers but full KOR agonists might lead to undesired side-effects. By ligation of a small molecule KOR ligand to peptides binding a secondary receptor site the authors aimed to obtain highly selective ligands with an improved pharmacological profile. Ligands were designed computationally and characterized in vitro and in vivo. The authors found that conjugate DNCP- β -NalA(1) shows partial agonism at KOR with bias towards G-protein signaling over arrestin-recruitment and were able to depict the binding position by cry-EM.

In the introduction, the authors explain the relevance of new opioid receptor ligands, especially targeting KOR. Problems of analgesic drugs targeting opioid receptors are pointed out and promising strategies to avoid severe side effects are presented. The authors explain their approach to design peptide-small molecule conjugates as KOR ligands and describe the basis of combining a KOR small molecule ligand with cyclic peptides.

In the results, it is well explained how peptide-small molecule conjugates are designed using computational methods and exploiting the recently published structure of human KOR bound to MP1104. Explanation of the design process is well supported by Figure 1, only the text in Fig.1 b(6) is too small. It becomes clear how peptides were selected for synthesis and characterized using binding assays prior ligation to the small molecule. In binding and functional cAMP assays of the conjugates, an enhanced efficacy and potency has been shown compared to the small molecule itself. A selected conjugate was further profiled regarding receptor subtype selectivity and functional bias. The authors refer to the used arrestin subtype as β -arrestin-2(3), selecting one of the nomenclatures would make it easier to understand. Usually arrestin-2 and -3 is preferred in non-beta-adrenergic systems. In vivo pharmacology of the conjugate was investigated in mice. Significant and potent antinociception in was shown, without KOR-mediated side effects. The predicted binding of the peptide-small molecule conjugate was checked by determination of the cryo-EM structure of human KOR bound to the conjugate, supported by mutagenesis data. In the discussion, the authors explain how they obtained a full KOR agonist with bias towards G-protein signaling over β -arrestin recruitment. Selectivity for KOR over MOR was achieved. As the conjugate is equally potent at MOR as at KOR, it should be further discussed if this could also account to the in vivo activity. As studies are also performed in mice, a short comparison of the human and rodent KOR would be helpful. The authors critically discuss, that their design was biased towards the used cryo-EM structure and that molecular dynamics simulation could improve the design protocol. This study shows that the approach of ligating an orthosteric binding small molecule with a peptide binding to a secondary binding site is a promising way to design selective ligands.

The methods section shows, that a broad range of methods was used in the study ranging from computational design over synthesis of the peptides and conjugates to in vitro and in vivo characterization. Cry-EM was used to validate the computational data. All procedures, including statistical data analysis are well explained.

Reviewer #2:

Remarks to the Author:

I've been asked to comment on the in vivo pharmacology studies in this manuscript describing new kappa-opioid receptor ligands, having no relevant expertise on structure modeling or screening. The data comprise dose-response curves for DNCP- β -NalA(1) and U50-488H (for comparison) on the formalin test of nociception and the rotarod test of sedation. The data are convincing, and fully support the conclusion that DNCP- β -NalA(1) produces dose-dependent analgesia without apparent sedation or confounding effects of edema, and in a nor-BNI-reversible manner. The measurement of sedation and edema is a plus. My only major criticism is one of scope. The formalin test and the formalin test alone is probably not enough to convince that a novel compound is a potentially clinically useful analgesic. Especially since the last sentence of the abstract claims that this work "may drive the development of...therapeutics for...chronic pain", the

additional use of an assay of chronic pain would be preferable. Finally, in 2023 I don't think there is any defensible reason to only use male mice in preclinical research.

Reviewer #3:

Remarks to the Author:

In this study, Muratspahić et al. described a computational approach to design cyclic peptide-small molecule conjugates targeting opioid receptor KOR based on the Rosetta peptide design software. Utilizing this approach, the authors firstly de novo designed and screened several thioether cyclic peptides potentially interacted with the ECL2/3 of KOR, as the interactions of ligand with ECL regions of GPCR were commonly associated with the ligand selectivity and efficacy. Conjugation of the de novo cyclic peptides (DNCP) with a β -naloxamine (β -NalA) moiety generated the DNCP- β -NalA (1-6) conjugates, among which the DNCP- β -NalA (1) occupied the highest affinity and low nanomolar potency. Investigation on the in vitro and in vivo pharmacological profiles of DNCP- β -NalA (1) indicated it to be selective KOR agonist with attenuated arrestin activity, and conducted KOR-mediated analgesic/anti-inflammatory effects with reduced side effects such as sedation in mice models. Finally, the authors validated the binding mode of designed conjugate DNCP- β -NalA (1) in KOR by cryo-EM structure determination of KOR-Gi signaling complex bound to DNCP- β -NalA(1). As expected, the cyclic peptide moiety interacted mainly with ECL2/3 of KOR. Mutagenesis studies suggested important role of the extracellular vestibule of KOR in controlling the ligand selectivity and efficacy, including ECL2/3 and extracellular ends of TM6/7.

The structure-based design of peptide drugs targeting on GPCRs has long been a challenge task in the field of GPCR drug discovery. This study stands out as a significant progress in the development of de novo computational approach for designing cyclic peptide- small molecule conjugates of GPCR with desired binding properties and pharmacological profiles such as ligand potency and efficacy, which may broadly be applied to the rational design of similar drugs of other GPCRs. Another strength of this study is the discovery and characterization of peptide- small molecule conjugate DNCP- β -NalA(1) as potent and G protein-biased KOR ligand, which provides valuable information about KOR pharmacology. However, there are a number of points need to be addressed prior to publication.

Major issues:

In the "Structural validation of peptide-small molecule design" part, the authors detailly analyzed the interactions of cyclic peptide moiety of DNCP- β -NalA(1) with KOR extracellular vestibule (Fig.5 and Fig. S17). However, the density of cyclic peptide moiety of DNCP- β -NalA(1) was poorly resolved, which is hard to support the detailed interactional information, as well as the binding pose of the cyclic peptide moiety. One would think that the weak density/ flexibility of the cyclic peptide moiety may be attributed to the weak interaction of the cyclic peptide with ECL2/3 of KOR. Since the cryo-EM structural information is important to validate the designed mode of the DNCP- β -NalA(1), the authors need to improve the quality of the ligand density in the DNCP- β -NalA(1) bound KOR structure, specifically in the cyclic peptide part. In addition, several regions of the current DNCP- β -NalA(1)-KOR structure model are not well fitted with the cryo-EM density, including β -NalA moiety of DNCP- β -NalA(1), as well as the ECL2, ECL3 and ICL3 loops of KOR. Thus, the structure model needs to be better refined.

Minor issues:

1. As indicated in the manuscript, the side chain of tertiary amino group in β -NalA moiety of DNCP- β -NalA(1) is allyl, rather than propyl as showed in the structure model and figures.
2. Fig.3b-3d: It is confused that the authors used different reference ligands to characterize the G protein and β -arrestin activities of DNCP- β -NalA(1) and β -NalA. Adding the endogenous peptide ligand dynorphin A as reference ligand in Fig.3b-3c or U69593 as reference ligand in Fig.3d will be helpful to better elucidate the potency and efficacy of DNCP- β -NalA(1) and β -NalA.
3. In the "Pharmacological profiling of DNCP- β -NalA (1) for receptor subtype selectivity and functional bias" section, the pharmacological data of DNCP- β -NalA(1) to NOPR, including those of radioligand binding assay and cAMP inhibition assay, will be necessary to better reveal the receptor subtype selectivity of DNCP- β -NalA(1).

4. Line 226: "SI Table S5 and S6" should be "SI Table S3".
5. As shown in the DNCP- β -NalA(1)-KOR structure solved here, residues E209ECL2, E2976.58 and L3097.32 all form extensive interactions with the ligand. However, mutations of these residues into alanine led to increased effects, instead of reduced effects, on potency or efficacy in G protein activation and arrestin recruitment. This should be clarified. Recent structural and functional work on KOR bound to dynorphin suggested important role of E209ECL2 and E2976.58 in the potency and efficacy of dynorphin. It will be encouraged to add more comparison and discussion on the opposite role of these residues in potency or efficacy of DNCP- β -NalA(1) and dynorphin, which may provide more insight into the fine-tuning of the KOR extracellular vestibule to ligand pharmacology.
6. Fig. S15: Please include the detailed cryo-EM data processing procedures and the local density maps of TM1-TM7, ECL2 and ECL3 of KOR, and the ligand DNCP- β -NalA(1).
7. The data of size exclusion chromatography and SDS-PAGE analysis of purified DNCP- β -NalA(1)-KOR-Gi should be provided.

Reviewer #4:

Remarks to the Author:

The manuscript 'De novo design and structural validation of peptide-drug conjugate ligands of the kappa-opioid receptor' described an approach for the design and development of a novel kappa-opioid receptor ligand by conjugating a cyclic hexapeptide to a small molecule β -naloxamine (NalA). The backbones of the cyclic peptides were generated using Rosetta peptide design method and the sequence of the hexapeptide was determined through the iterative in silico optimization of the interactions with the KOR orthosteric binding pocket, as well as its extracellular loops (ECL2 and/or ECL3). Using these methods, the authors identified a cyclic peptide-small molecule conjugation DNCP- β -NalA(1) as a potent agonist at KOR. Compared with the small molecule β -NalA alone, DNCP- β -NalA(1) exhibited a much-improved binding affinity (3.9 nM vs 72 nM), agonist potency (5.5 nM vs. 130 nM), and efficacy (maximum % stimulation: 85% vs 57%, compared to U69,593) at the KOR. Though DNCP- β -NalA(1) forms additional interactions with KOR at ECL2 and ECL3, it showed slightly less β -arrestins recruitment compared to β -NalA alone. Subcutaneous (s.c.) administration of DNCP- β -NalA(1) produced a dose-dependent antinociceptive effect in the mouse model of formalin-induced acute inflammatory pain with an ED50 value of 1.64 μ mol kg⁻¹ (95% CL, 0.76-3.53). The compound did not produce significant KOR-mediated motor dysfunction/sedation in mice at a high dose of 7.6 μ mol kg⁻¹ (s.c). The authors also determined the cryo-EM structure of human KOR bound to DNCP- β -NalA(1) and Gai1/Gb1/Gy2 heterotrimer at a map resolution of 2.6 Å, and compared the computationally designed model with the cryo-EM structure. However, the peptide conformation and its binding pose at KOR were not fully validated by structural determination using cryo-EM.

Overall, this manuscript reported a novel idea/approach for the design and development of new ligands at KOR. This method may be applied to the design of other ligands at other receptors. The manuscript presents a large amount of data including computational design, pharmacological evaluation, and cryo-EM structure determination. However, some of the data needs clarification, more detailed interpretation, and more vigorous validation. The manuscript at this stage is not appropriate for publication in Nature Communication. The major points are listed below.

1. Lines 55-59, The authors stated 'Small molecules are a common therapeutic modality for targeting GPCRs² due to their low cost, high stability, lipophilicity and oral bioavailability; however, they often have limited target selectivity, which can result in undesired off-target effects, and adverse clinical events^{3,4}. A prominent example is the ongoing and rapidly evolving global opioid crisis accompanied by substantial opioid-related morbidity and mortality^{5,6}. Prescribed opioid analgesics including fentanyl, morphine and their derivatives, that act primarily via the mu-opioid receptor (MOR), have numerous and serious side effects⁷.' —It is odd to link 'limited target selectivity' to 'undesired off-target effects.'
2. Lines 108-112, 'At the position directly conjugated to the β -NalA, we placed a D-phenylalanine to mimic the MP1104 iodobenzamide group, and at the other position we sampled all 20 amino acids (excluding glycine and cysteine) in L and D forms, aiming for interactions with the extracellular loops of the receptor (SI Fig. S1c/d). Over all combinations of backbone conformations and amino acid choices we chose four solutions with the lowest Rosetta binding energy for KOR.' —(1) The Authors claim β -NalA was used to design their molecules, however, the

S1 Fig.S1c shows that the docked structure is a part of the morphinan structure of MP1104, not β -NalA. (2) The authors should clarify what '4 solutions' are and provide data to support their choice of the 4 solutions.

3. In Fig.2b, the six cyclic hexamers show the different backbone shapes. Do they interact with the same set of residues at ECL2/3? The authors need to add a table in Fig 2 below the general chemical structure of DNCP- β -NalA(1-6) to show the compositions of DNCP- β -NalA(1-6).

4. Lines 171-172, 'We obtained four DNCP- β -NalA conjugates (1-4) (SI Table S1); the conjugation of DNCP (35) and DNCP (36) was unsuccessful (SI Fig. S9 and Table S1).' —SI Table S1 have no data to support the claim.

5. Lines 173-174, Fig 3a&b and Table S3. —Table S3 shows DNCP- β -NalA (2) produced agonist activity at mKOR with an EC50 of 7.5 nM in cAMP assay, while its binding affinity being 31 nM. Also, DNCP- β -NalA (4) showed an EC50 of 1 nM and its binding affinity 31 nM at mKOR. What may cause their binding affinity values 4-to-10X lower than their EC50 values?

6. Lines 189-190, 'DNCP- β -NalA(1) fully activated human KOR (EC50 = 5.5 nM; Emax = 85%) compared to the partial agonist β -NalA (EC50 = 130 nM; Emax = 57%) and the reference KOR agonist U69,593 (Fig. 3c and SI Table S4).' —Fig. 3c does not indicate β -NalA could have an Emax = 57%.

7. Lines 241-242, 'Herein, DNCP- β -NalA(1) bound to mouse MOR and DOR with Ki values of 5.4 nM and 318 nM, respectively, supporting an ~80-fold selectivity for KOR over DOR (Fig. 3f).' — While this is true, the authors should also point out that DNCP- β -NalA(1) did not show selectivity at MOR vs. KOR. This is a mixed MOR/KOR ligand, but there is no functional activity data at MOR. It is important to characterize its functional activity at MOR.

8. Lines 248-250, 'Systemic subcutaneous (s.c.) administration of DNCP- β -NalA(1) produced a dose-dependent reduction in the pain behavior of formalin-injected mice with significant effects at doses of 1.9 and 3.8 μ mol kg⁻¹ (Fig. 4a).' —Was the peptide bond connecting Thr and DNCP stable in vivo? Is there any data, in vitro and in vivo, suggesting that Thr-D-Phe- β -NalA did not activate KOR/MOR/DOR and produce antinociception?

9. Line 288, 'U50,488 did not affect paw oedema formation at any of the tested doses when compared to the saline group.' —U50,488 was reported to produce anti-inflammatory effects in a chronic inflammatory rodent model (Rheumatology, 2006 (45) 295-302). The authors should explain this difference.

10. Lines 321-323, 'The initial goal of the design was to form interactions with both the ECL2 and ECL3. We are still seeing these interactions but in an altered manner: in the designed model the peptide was predicted to have a D-Tyr (DCNP1_design D-Tyr-R3) interaction with ECL2 and a Tyr (DCNP1_design Tyr-R1) interaction with ECL3. We found that the residue D-Tyr (R3) interacted with the ECL3 versus the predicted interaction with ECL2 (SI Fig. S3).' —The authors claim that based on the computational design, they only selected to synthesize 6 compounds but end up with a potent KOR agonist DNCP- β -NalA(1). However, the docking pose of DNCP1 looked very different from the cryo-EM structure. Is the selection of the six compounds pure luck?

11. Lines 330-331, 'It is interesting that D1383.32 appears to be too far to form the H-bond interaction with DNCP- β -NalA(1) and the D1383.32N mutation caused only an 8-fold loss of potency for DNCP- β -NalA(1), but a 1,000-fold for U50,488'. — (1) If the cryo-EM structure data shows DNCP- β -NalA(1) does not form HB with D138, which is considered a critical residue/interaction utilized for the computational design of DNCP- β -NalA(1). How reliable would the results of the computational design be? (2) Fig. S14 is important to support this claim, should be moved to and combined with Fig.5. (3) Does U50,488 bind to KOR in the same binding pocket(s) using same sets of residues as does DNCP- β -NalA, or as shown in cyro-EM structure?

12. Lines 338-340, 'Given that pentazocine is a partial agonist at KOR (Emax: 40% of U50,488) and adopts a similar binding pose as the small-molecule portion of DNCP- β -NalA(1), this suggests that specific residues in the orthosteric pocket are sufficient to regulate ligand efficacy, as supported by I1353.29A and K2275.39A that cause a loss of 50% efficacy compared to the wild type'. —It is not clear what the authors want to claim here. Is interaction with D138 critical or not that important in terms of receptor binding and ligand function?

13. Lines 344-345, 'Several residue mutations, such as E209ECL2A, E2976.58A, and L3097.32A led to increased potency in G protein activation and enhanced efficacy in arrestin recruitment.' — Do these residues E209, E297, and L309 interact with DNCP- β -NalA(1) in the cryo-EM structure? How can this information be translated to guide the computational design of a more potent KOR agonist?

14. Lines 366-369, 'we overcome multiple rounds of structure-based design and pharmacological

testing; we needed to synthesize and experimentally characterize only 4 compounds to discover a high-affinity molecule with novel patterns of GPCR signaling and pharmacology.' —As the docking pose is not validated by cryo-EM structure, it seems the authors are lucky to discover DNCP- β -NaIA(1) with only synthesis of 4 compounds. Also, since DNCP- β -NaIA(1) exhibited binding at MOR as potent as at KOR, and the functional activity at MOR is not characterized, its GPCR signaling and pharmacology are not well characterized.

Some minor points also need be clarified.

15. Lines 100, 'The internal cycle reduces flexibility and thus reduces the entropy loss upon binding which can increase binding affinity and enhance stability^{4,28}.' —What does 'the internal cycle' mean?

16. Line 103, 'we chose to employ the cyclic component of the lariat 5-6 residues closed by thioether macrocyclization linking a Cys side chain and the N-terminus'. —Add (Fig. 1b) to the end of the sentence.

17. In Fig 1b (3), what is the structure showing in ball & stick style?

18. SI Fig S8, add calc. [M+H]⁺ to each structure.

#	Comments	Response	Page #
Reviewer 1			
1	Explanation of the design process is well supported by Figure 1, only the text in Fig.1 b(6) is too small.	Thank you for the comment on the design process. We have updated the plots with simplified labeling and thus increased font size. Please note that a more comprehensive filtering scheme can be found in Fig. S2 (Histograms of select metrics of the output designs for 6-mers). The values in this Figure are visible and clear.	p. 5 p. S3
2	The authors refer to the used arrestin subtype as β -arrestin-2(3), selecting one of the nomenclatures would make it easier to understand. Usually arrestin-2 and -3 is preferred in non-beta-adrenergic systems.	Accordingly, this has been clarified and the nomenclature has been revised throughout the manuscript to β -arrestin-1 and -2.	throughout
3	In the discussion, the authors explain how they obtained a full KOR agonist with bias towards G-protein signalling over β -arrestin recruitment. Selectivity for KOR over DOR was achieved. As the conjugate is equally potent at MOR as at KOR, it should be further discussed if this could also account to the in vivo activity.	We have now added more detailed in vitro pharmacology data supporting the competitive antagonism of DNCP- β -NalA(1) at MOR (Schild regression analysis using the cAMP and [³⁵ S]GTP γ S assays with DAMGO as reference MOR agonist, see SI Fig. S13 and Fig. S14). In addition we determined the affinity of DNCP- β -NalA(1) to the nociceptin (NOP) receptor using a radioligand binding assay (see SI Fig. S12). Further, we discussed the contribution of DNCP- β -NalA(1)'s competitive antagonism at MOR to its in vivo activity: " The mixed action of DNCP-β-NalA(1) as biased agonist at KOR and competitive antagonist at MOR might thus contribute to the observed antinociceptive efficacy and result in a favorable side effect profile. The development of mixed KOR agonists/MOR antagonists has been explored as a strategy to develop safer pain medications (Dalefield et al, Front. Pharmacol., 2022) ".	p. 2 p. 9/10, p. S14- S16 p. 16 (lines 630-633)
4	As studies are also performed in mice, a short comparison of the human and rodent KOR would be helpful.	Thanks for the suggestion. Accordingly, we have added a sequence alignment of human and mouse KOR (SI Fig. S10).	p. S12
Reviewer 2			
1	The data are convincing, and fully support the conclusion that DNCP- β -NalA(1) produces dose-dependent analgesia without apparent sedation or confounding effects of edema, and in a nor-BNI-reversible manner. The measurement of sedation and edema is a plus. My only major criticism is one of scope. The formalin test and the formalin test alone is probably not enough to convince that a novel compound is a potentially clinically useful analgesic. Especially since the last sentence of the abstract claims that this work "may drive the development of...therapeutics for...chronic pain", the additional use of an assay of chronic pain would be preferable.	Thanks for the comment on the convincing nature of the data. As suggested, we have included in vivo data demonstrating the antinociceptive efficacy of DNCP- β -NalA(1) in a mouse model of chronic pain, i.e. Complete Freund's Adjuvant (CFA)-induced inflammatory hyperalgesia. We demonstrate a dose- and time-dependent inhibition of pain response to thermal stimulation (see Fig. 4d and e). We also demonstrated that the antinociceptive effect of DNCP- β -NalA(1) in experimental chronic pain is KOR-dependent (see Fig. 4f). The new data support our original claim in the abstract. This has been addressed in the text: " Next, we investigated the antinociceptive efficacy of DNCP-β-NalA(1) after s.c. administration in a mouse model of chronic inflammatory pain. Chronic pain was induced by injection of Complete Freund's Adjuvant (CFA) to the dorsal side of the right hindpaw, evidenced by a significant reduction at 72 h post-inoculation versus pre-innoculation ($P < 0.001$, paired t-test) in paw withdrawal thresholds to thermal stimulation assessed with the Hargreaves test. Mice were treated s.c. with saline, and different doses of DNCP-β-NalA(1), or U50,488, and tested for thermal sensitivity (Fig. 4d and 4e, respectively). DNCP-β-NalA(1) produced time- and dose-dependent increase in the inflamed paw withdrawal latencies. Compared to saline-treated mice, DNCP-β-NalA(1) significantly reduced thermal sensitivity at doses of 0.8 and 1.9 $\mu\text{mol kg}^{-1}$ (Fig. 4d). Notable was the fast onset of the antihyperalgesic effect of DNCP-β-NalA(1) with a peak effect at 15 min followed by a rapid decline, with thermal nociceptive thresholds returning to basal values at 2 h after drug administration. Administration of U50,488 also caused a dose-dependent attenuation in pain behavioral of mice with CFA-	p. 10-12 (lines 381-469)

		induced inflammatory hyperalgesia (Fig. 4e). Doses of 0.6 and 2.1 mg kg⁻¹ of U50,488 significantly increased paw withdrawal latencies from 15 min to 1 h, and from 15 min to 6 h, respectively, with a peak antinociceptive effect at 30 min. Although DNCP-β-NalA(1) had a shorter duration of the antinociceptive effect than U50,488, it showed comparable antinociceptive efficacy at the highest tested dose in attenuating the pain response in mice with CFA-induced inflammatory hyperalgesia. We also demonstrated that the antinociceptive effect of DNCP-β-NalA(1) (1.9 mg kg⁻¹) was reversed by pretreatment with the nor-BNI (13.6 μmol kg⁻¹, s.c.), indicating that the KOR is involved in DNCP-β-NalA(1) in vivo agonist activity (Fig. 4f). Altogether, these data show that DNCP-β-NalA(1) efficiently reversed thermal hyperalgesia in mice with CFA-induced chronic inflammatory pain acting through the KOR, with a fast onset of action."	
2	In 2023 I don't think there is any defensible reason to only use male mice in preclinical research.	We agree, and we will generally consider the use of animals of both sexes for in vivo studies. However, in our past experience male animals are better suited in this specific model to exclude any potential influence on the behavioural outcome due to the menstrual cycle in female animals; this has been common practice in antinociceptive drug discovery.	-
Reviewer 3			
1	This study stands out as a significant progress in the development of de novo computational approach for designing cyclic peptide-small molecule conjugates of GPCR with desired binding properties and pharmacological profiles such as ligand potency and efficacy, which may broadly be applied to the rational design of similar drugs of other GPCRs. Another strength of this study is the discovery and characterization of peptide- small molecule conjugate DNCP-β-NalA(1) as potent and G protein-biased KOR ligand, which provides valuable information about KOR pharmacology.	Thank you very much for highlighting the significance and novelty of our study. Very much appreciated!	-
2	In the "Structural validation of peptide–small molecule design" part, the authors detailedly analyzed the interactions of cyclic peptide moiety of DNCP-β-NalA(1) with KOR extracellular vestibule (Fig.5 and Fig. S17). However, the density of cyclic peptide moiety of DNCP-β-NalA(1) was poorly resolved, which is hard to support the detailed interactional information, as well as the binding pose of the cyclic peptide moiety. One would think that the weak density/ flexibility of the cyclic peptide moiety may be attributed to the weak interaction of the cyclic peptide with ECL2/3 of KOR. Since the cryo-EM structural information is important to validate the designed mode of the DNCP-β-NalA(1), the authors need to improve the quality of the ligand density in the DNCP-β-NalA(1) bound KOR structure, specifically in the cyclic peptide part. In addition, several	We would like to thank the reviewer for the advice to improve the structural model. We have now further improved the KOR-DNCP-β-NalA(1) model by generating a local refined map of the receptor region. The final model is built based on the density from the global B-factor sharpened map and a deep enhanced sharpened map. For example, the density of the ligand or side chains of the residues may be poor in one map but show improved density in another, we built the model based on the map that has the greatest density. To specifically answer the questions of the reviewer: (i) The β-NalA moiety of the ligand has been improved, as shown in updated Fig 5. (ii) The cyclic peptide moiety has incomplete density in both maps likely due to the weak interactions and flexibility. To support the observed binding poses and specific interactions, we performed molecular dynamic simulations to examine distance of ligand and interacting residues. As shown in SI Fig. S23, the simulations' reliability was assessed by analyzing the root mean square deviation of protein backbone atoms indicating consistent stability throughout. We then examined the residues involved in overall ligand-receptor interactions. We calculated the fraction of protein	p. 13/15 p. S25, p. S26, p. S30, p. S19

	regions of the current DNCP-β-NalA(1)-KOR structure model are not well fitted with the cryo-EM density, including β-NalA moiety of DNCP-β-NalA(1), as well as the ECL2, ECL3 and ICL3 loops of KOR. Thus, the structure model needs to be better refined.	amino acid residues within a 6.0 Å distance from the bound ligand in the combined simulations trajectories (SI Fig. S24). Within the small-molecule binding pocket 1, the ligand demonstrated molecular interactions with Q115^{2,60}, L135^{3,29}, Y139^{3,33}, M142^{3,36}, V230^{5,42}, W287^{6,48}, I290^{6,51}, I316^{7,39}, G319^{7,42}, and Y320^{7,43}, which align well with the interactions observed in the previously published structure (PDB: 6B73). In peptide-ring binding pocket 2, the ligand primarily engaged in molecular interactions with E209^{ECL2}, C210^{ECL2}, L212^{ECL2}, and Y313^{7,36}, as well as amino acid residues S303^{ECL3}, H304^{ECL3}, A308^{7,31}, L309^{7,32} in strong agreement with the cryo-EM data. The peptide portion of the ligand also had significant contact frequency with amino acid residues located on TM6 and TM7: F293^{6,54}, I294^{6,55}, E297^{6,58} and Y312^{7,35} (SI Fig. S24). Using MD simulations, we proceeded to analyze the interactions between the ligand and specific amino acid residues (D138, E209, E297 & L309) observed in the cryo-EM structure (SI Fig. S28). The simulations suggest potential interactions consistent with what we observed as the highest distance probability was consistently less than 6Å in most simulations. In certain simulations, we noticed larger values in the distance distribution probability, indicating increased flexibility between the residues. This higher flexibility likely contributes to the poor density of the peptide part compared to the small molecule part. Notably, the MD simulations revealed a broad distance distribution between the ligand and D138, suggesting weaker molecular interactions with the ligand. This is consistent with our functional studies that the mutation of D138 to N138 has a less deleterious effect on DNCP-β-NalA (1) compared to that on U50,488 or endogenous ligand dynorphin A 1-13. (iii) We have re-examined the model by deleting the regions that show no density of fragmented density in the map, particularly the N terminal, ECL2, ECL3, and ICL3 (SI Fig. S17).	
3	As indicated in the manuscript, the side chain of tertiary amino group in β-NalA moiety of DNCP-β-NalA(1) is allyl, rather than propyl as showed in the structure model and figures.	We thank the reviewer for pointing out this error. We have corrected it to allyl in the figures. This error was because that Pymol did not show the double bond correctly.	-
4	Fig.3b-3d: It is confused that the authors used different reference ligands to characterize the G protein and β-arrestin activities of DNCP-β-NalA(1) and β-NalA. Adding the endogenous peptide ligand dynorphin A as reference ligand in Fig.3b-3c or U69593 as reference ligand in Fig.3d will be helpful to better elucidate the potency and efficacy of DNCP-β-NalA(1) and β-NalA.	We have added functional data on the reference ligand dynorphin A of G protein activation in the cAMP and [³⁵S]GTPγS binding assay (see Fig. 3b,c, SI Table S4) as well as U69,593 in β-arrestin-2 recruitment assays (Fig. 3d, SI Table S3).	p. 8 p. S34/35
5	In the “Pharmacological profiling of DNCP-β-NalA (1) for receptor subtype selectivity and functional bias” section, the pharmacological data of DNCP-β-NalA(1) to NOPR, including those of radioligand binding assay and cAMP inhibition assay, will be necessary to better reveal the receptor subtype selectivity of DNCP-β-NalA(1).	As suggested by the reviewer, we tested the binding affinity of DNCP-β-NalA(1) for NOP receptor in a radioligand binding assay. DNCP-β-NalA(1) has a very low binding affinity at the NOP receptor with a Ki value of >1 μM, being more than 300-fold selective for KOR vs. NOP receptor (SI Fig. S12).	p. 9/10 and p. S14

6	Line 226: "SI Table S5 and S6" should be "SI Table S3".	This has been corrected.	p. 9
7	As shown in the DNCP-β-NalA(1)-KOR structure solved here, residues E209^{ECL2}, E2976.58 and L3097.32 all form extensive interactions with the ligand. However, mutations of these residues into alanine led to increased effects, instead of reduced effects, on potency or efficacy in G protein activation and arrestin recruitment. This should be clarified. Recent structural and functional work on KOR bound to dynorphin suggested important role of E209^{ECL2} and E2976.58 in the potency and efficacy of dynorphin. It will be encouraged to add more comparison and discussion on the opposite role of these residues in potency or efficacy of DNCP-β-NalA(1) and dynorphin, which may provide more insight into the fine-tuning of the KOR extracellular vestibule to ligand pharmacology.	We would like to thank the reviewer for the advice. It is indeed interesting to see that mutations of several residues in the ECL2/3 can further increase the potency and efficacy of DNCP-β-NalA(1). As the extracellular portions of GPCRs undergo dynamic conformational displacement upon receptor activation. We argue that the gain of functions is likely due to the removal of steric clashes when mutating those residues. Consistent with recent reported structures of opioid receptors bound to different endogenous peptides (Wang et al., Cell 2022), our structure and functional results suggest that ECL2/3 and extracellular transmembrane ends play important roles in regulating the efficacy of opioid receptor activation. It is worth pointing out that, although only the first 8 residues of dynorphin A₁₋₁₃ has been modelled, the binding of DNCP-β-NalA(1) shows some overlapping with dynorphin's binding pose (SI Fig. S29). Future ligand design should consider targeting these less conserved regions to achieve ligand selectivity and efficacy. We have added this to the manuscript as part of the discussion. Figure R6. Comparison of binding poses between DNCP-β-NalA(1) and dynorphin A₁₋₁₃. The binding pose of dynorphin is adopted from PDB ID: 8F7W (Wang et al., Cell 2022). We followed the reviewer's advice and compared DNCP-β-NalA(1) with endogenous dynorphin A₁₋₁₃ in gain-of-function mutants. Again, E209^{ECL2A}, E2976.58A and L3097.32A all led to increased potency and efficacy of DNCP-β-NalA(1) as we reported, but only E209^{ECL2A} and E2976.58A significantly reduced the potency of dynorphin A₁₋₁₃ by 3.4-, 4.7-fold, respectively, and all mutations didn't affect the efficacy of dynorphin A₁₋₁₃ (SI Fig. S29, Tables S13 and S14). This difference may arise from the dynamics of extracellular vestibules that display specificity upon different ligand binding. We have added this to manuscript and supplemental information: "The more profound observation was from the mutational analysis of residues in the second binding pocket (Fig. 5d) that may have formed H-bond or hydrophobic interactions (residues within 4 Å of the ligand) with the peptide ring. Several residue mutations, such as E209^{ECL2A}, E2976.58A and L3097.32A led to increased potency in G protein activation and enhanced efficacy in arrestin recruitment (SI Fig. S29, SI Table S12 and SI Table S13). The role of extracellular vestibule has been investigated in the KOR-dynorphin structure³⁸, in which mutations of ECL2/3 led to significant reduction of dynorphin A₁₋₁₃'s agonist activity. Comparison of dynorphin with DNCP-β-NalA(1) shows overlapping of several binding sites (SI Fig. S29). Characterization of dynorphin in KOR E209^{ECL2A}, E2976.58A or L3097.32A mutants displayed opposite effects compared to DNCP-β-NalA(1), suggesting that the ECL2/3 and extracellular transmembrane ends play important roles in regulating ligand-specific responses during opioid receptor activation (SI Fig. S29, SI Table S13 and SI Table S14). The MD simulations indicate potential interactions between the ligand and amino acid residues (E209^{ECL2}, E2976.52 and L3097.32). In most simulations, the highest probability of distance consistently remained below 6 Å. However, in certain simulations, larger distance probability values were observed, indicating increased flexibility between the ligand and those specific residues (SI Fig. S28)."	p. 15 (lines 576-591) p. S31, p. S45/46

8	Fig. S15: Please include the detailed cryo-EM data processing procedures and the local density maps of TM1-TM7, ECL2 and ECL3 of KOR, and the ligand DNCP- β -NalA(1).	We thank the reviewer for pointing out this missing information. We have included the cryo-EM data processing procedures and local density maps of TM1-TM7, ECL2 and ECL3 of KOR, and ligand DNCP- β -NalA(1) (SI Fig. S17)	p. S19
9	The data of size exclusion chromatography and SDS-PAGE analysis of purified DNCP- β -NalA(1)-KOR-Gi should be provided.	We thank the reviewer for this great suggestion. We have included the size exclusion chromatography data of DNCP- β -NalA(1)-KOR-Gi1-G γ 2-G β 1-Scfv16 complex purified on Superdex 200 Increase (10/300) column and data of Coomassie brilliant blue-stained SDS-PAGE analysis of the sample for making cryo-electron microscopy (cryo-EM) grids (SI Fig. S17).	p. S19
Reviewer 4			
1	The authors also determined the cryo-EM structure of human KOR bound to DNCP- β -NalA(1) and Gai1/Gb1/G γ 2 heterotrimer at a map resolution of 2.6 Å, and compared the computationally designed model with the cryo-EM structure. However, the peptide conformation and its binding pose at KOR were not fully validated by structural determination using cryo-EM.	We would like to thank the reviewer for the advice to improve the model. We have now further improved the KOR-DNCP- β -NalA(1) model by generating a local refined map of the receptor region. The final model is built based on the density from the global B-factor sharpened map and a deep enhanced sharpened map. For example, the density of the ligand or side chains of the residues may be poor in one map but show improved density in another, we built the model based on the map that has density. To support the incomplete density of the cyclic portion, we also performed MD simulations to support our structural observations, as shown below (and Response to reviewer 3).	-
2	Overall, this manuscript reported a novel idea/approach for the design and development of new ligands at KOR. This method may be applied to the design of other ligands at other receptors. The manuscript presents a large amount of data including computational design, pharmacological evaluation, and cryo-EM structure determination. However, some of the data needs clarification, more detailed interpretation, and more vigorous validation. Lines 55-59, The authors stated ' Small molecules are a common therapeutic modality for targeting GPCRs2 due to their low cost, high stability, lipophilicity and oral bioavailability; however, they often have limited target selectivity, which can result in undesired off-target effects, and adverse clinical events. A prominent example is the ongoing and rapidly evolving global opioid crisis accompanied by substantial opioid-related morbidity and mortality. Prescribed opioid analgesics including fentanyl, morphine and their derivatives, that act primarily via the mu-opioid receptor (MOR), have numerous and serious side effects7. ' - It is odd to link 'limited target selectivity' to 'undesired off-target effects.	We appreciate the positive feedback to our manuscript and have addressed all constructive criticism to improve the data. As suggested, we corrected this statement: " Small molecules are a common therapeutic modality for targeting GPCRs due to their low cost, high stability, lipophilicity and oral bioavailability; however, they often have limited target selectivity and are associated with off-target effects and adverse clinical events (Muratspahić et al., TiPS, 2019; Muttenthaler et al., Nat Rev Drug Discov, 2021). A prominent example is the ongoing and rapidly evolving global opioid crisis accompanied by substantial opioid-related morbidity and mortality. Prescribed opioid analgesics including fentanyl, morphine and their derivatives, that act primarily via the mu-opioid receptor (MOR), have numerous and serious side effects (Del Vecchio et al., ACS Chem Neurosci, 2017) ".	p. 3 (lines 74-80)
3	Lines 108-112, 'At the position directly conjugated to the β -NalA, we placed a D-phenylalanine to mimic the MP1104 iodobenzamide group, and at the other position we sampled all 20 amino acids	We carefully revised the design section and noticed that we named MP1104 lacking iodobenzamide group β -NalA instead of N-cyclopropylmethyl-epoxy morphinan. This has accordingly been corrected.	p. 4 (lines 129-136)

	(excluding glycine and cysteine) in L and D forms, aiming for interactions with the extracellular loops of the receptor (SI Fig. S1c/d). Over all combinations of backbone conformations and amino acid choices we chose four solutions with the lowest Rosetta binding energy for KOR.’ -(1) The Authors claim β-NalA was used to design their molecules, however, the S1 Fig.S1c shows that the docked structure is a part of the morphinan structure of MP1104, not β-NalA. (2) The authors should clarify what ‘4 solutions’ are and provide data to support their choice of the 4 solutions.	“We started from a variant of MP1104 lacking the iodobenzamide group — N-cyclopropylmethyl-epoxy morphinan — bound to the KOR structure in the same orientation as MP1104. This MP1104 derivative has a free amine, which can be conjugated to the C-terminus of a peptide lariat. We first modeled two amino acid residues extending off the free amino group of the MP1104 derivative, and extensively sampled their backbone torsion angles. Next to the free amino group we placed a D-phenylalanine to mimic the MP1104 iodobenzamide group, and at the second position we sampled all 20 amino acids (excluding glycine and cysteine) in L and D forms, aiming for interactions with the extracellular loops of the receptor (SI Fig. S1c & d).” We also clarified the synthesis section and justified why we decided to use β-NalA instead of N-cyclopropylmethyl-epoxy morphinan as template for synthesis: “For synthesis of the of peptide–drug conjugates we chose β-NalA, distinguished by a less rigid morphinan structure and featuring an N-17 allyl group over the N-cyclopropylmethyl group of the MP1104 derivative (i.e. N-cyclopropylmethyl-epoxy morphinan), which was utilized during computational design. This decision was guided by the closely resembling core structure, along with the benefits of β-NalA’s simpler synthesis and a more flexible morphinan ring, thereby allowing the peptide region to bind the ECL2 region of KOR more tightly²⁹.” In addition, we clarified the four choices: “Over all combinations of backbone conformations and amino acid possibilities we chose four solutions (i.e. dipeptides D-Phe-L-Thr; D-Phe-L-Ser; D-Phe-L-Gln and D-Phe-L-Ala) with the lowest Rosetta binding energy for KOR that were next used to graft 5- and 6-mers thioether cyclized peptides onto.”	p. 7 (lines 213-219) p.4 (lines 136-139)
4	In Fig.2b, the six cyclic hexamers show the different backbone shapes. Do they interact with the same set of residues at ECL2/3? The authors need to add a table in Fig 2 below the general chemical structure of DNCP-β-NalA(1-6) to show the compositions of DNCP-β-NalA(1-6).	We now included a new SI Fig. S21 showing the interactions of DNCP31-36 with KOR. In addition, we generated a Table indicating whether computational models interact with ECL2/3 or not (SI Table S8). This has been described in the text: “We are still seeing these interactions but in an altered manner: in the designed model the peptide was predicted to have a D-Tyr (DNCP1_design D-Tyr–R₃) interaction with ECL2 and a Tyr (DNCP1_design Tyr–R₁) interaction with ECL3. We found that the residue D-Tyr (R₃) interacted with the ECL3 versus the predicted interaction with ECL2 (SI Fig. S21). Further structural information of the other computationally designed peptides whose ECL2/3 interactions are similar to those of computational DNCP-β-NalA(1) (SI Fig. S22) would elucidate the peptides’ ability to induce an altered conformation upon binding.” The Table with structures are already provided in Fig. 2. To clarify further, we added the following sentence in the caption of Fig. 2: “The amino acids indicated by R_# in Fig. 2b correspond to the identical side chain represented by R_# in Fig. 2c”.	p. 13 (lines 522 ff), p. S23, p. S40 p. 6
5	Lines 171-172, ‘We obtained four DNCP-β-NalA conjugates (1-4) (SI Table S1); the conjugation of DNCP (35) and DNCP (36) was unsuccessful (SI Fig. S9 and Table S1).’ SI Table S1 have no data to support the claim.	This has been corrected: “We obtained four DNCP-β-NalA conjugates (1-4) (SI Fig. S8 and Table S2)”.	p. 7 (lines 221-222)
6	Lines 173-174, Fig 3a&b and Table S3. Table S3 shows DNCP-β-NalA (2) produced agonist activity at mKOR with an EC50 of 7.5 nM in cAMP	We discussed what might have caused the discrepancy between potency and affinity values of the two conjugates: “Receptor reserve may account for this discrepancy between potency and affinity values of DNCP-β-NalA(2) and DNCP-β-	p. 7 (lines 219-222)

	assay, while its binding affinity being 31 nM. Also, DNCP- β -NalA (4) showed an EC50 of 1 nM and its binding affinity 31 nM at mKOR. What may cause their binding affinity values 4-to-10X lower than their EC50 values?	NalA(4) which can be observed in functional GPCR assay with opioid receptors (Kelly et al., Br J Pharmacol, 2013) ”.	
7	Lines 189-190, ‘DNCP- β -NalA(1) fully activated human KOR (EC50 = 5.5 nM; Emax = 85%) compared to the partial agonist β -NalA (EC50 = 130 nM; Emax = 57%) and the reference KOR agonist U69,593 (Fig. 3c and SI Table S4).’ Fig. 3c does not indicate β -NalA could have an Emax = 57%.	We carefully re-analyzed cAMP data of β -NalA. Indeed, the E _{max} of β -NalA is 61 ± 7 %. This has been corrected in the text and Table S3 .	p. 7 (line 230), p. S34
8	Lines 241-242, ‘Herein, DNCP- β -NalA(1) bound to mouse MOR and DOR with Ki values of 5.4 nM and 318 nM, respectively, supporting an ~80-fold selectivity for KOR over DOR (Fig. 3f).’ While this is true, the authors should also point out that DNCP- β -NalA(1) did not show selectivity at MOR vs. KOR. This is a mixed MOR/KOR ligand, but there is no functional activity data at MOR. It is important to characterize its functional activity at MOR.	As suggested, we characterized functional activity of DNCP- β -NalA(1) at MOR and demonstrated its potent competitive antagonism at mouse and human MOR in cAMP and [³⁵ S]GTP γ S binding assays, respectively: “ We next determined the mechanism of antagonism of DNCP-β-NalA(1) by measuring adenylyl cyclase-mediated cAMP inhibition and [³⁵S]GTPγS binding at mouse and human MOR, respectively, using Schild regression analysis. MOR expressed in HEK293 and CHO cells was activated by DAMGO in the absence and presence of increasing concentrations of DNCP-β-NalA(1). We observed a rightward shift of the concentration-response curves of DAMGO in cAMP (SI Fig. S14a) and [³⁵S]GTPγS binding assay (SI Fig. S14b). Schild analysis of DNCP-β-NalA(1) exhibited linear regression slopes of 0.9 and 1.5 (SI Fig. S14c, d) and pA2 values of 9.1 and 7.9 in cAMP and [³⁵S]GTPγS binding assays, respectively, which corresponds to an average functional affinity of 794 pM and 13 nM, respectively, thus demonstrating competitive antagonism of the DNCP-β-NalA(1) at MOR. ”.	p. 9-10 (lines 321ff)
9	Lines 248-250, ‘Systemic subcutaneous (s.c.) administration of DNCP- β -NalA(1) produced a dose-dependent reduction in the pain behavior of formalin-injected mice with significant effects at doses of 1.9 and 3.8 μ mol kg ⁻¹ (Fig. 4a).’ Was the peptide bond connecting Thr and DNCP stable in vivo? Is there any data, in vitro and in vivo, suggesting that Thr-D-Phe- β -NalA did not activate KOR/MOR/DOR and produce antinociception?	We thank the reviewer for this comment. To support the in vivo application of DNCP- β -NalA(1) we tested its serum stability in vitro and observed that DNCP- β -NalA(1) is stable in serum for at least 50 h (SI Fig. S15). This observation indicates that the conjugate is stable enough to elicit antinociception in mice.	p. 10 (line 367) p. S17
10	Line 288, ‘U50,488 did not affect paw oedema formation at any of the tested doses when compared to the saline group.’ U50,488 was reported to produce anti-inflammatory effects in a chronic inflammatory rodent model (Rheumatology, 2006 (45) 295-302). The authors should explain this difference.	We thank the reviewer for the recommended reference. We can comment on several possible explanations for the observed difference: (i) we measured the anti-inflammatory effect in a model of acute inflammatory pain, whereas the referred study is on chronic arthritic pain; (ii) we administered U50,488 systemically s.c. , whereas local contralateral administration, directly into the paw, was used in the referred study, accounting also for the difference in drug dose; and (ii) we tested U50,488 after single s.c. administration, whereas a repeated administration protocol was used in the referred study.	-

11	Lines 321-323, 'The initial goal of the design was to form interactions with both the ECL2 and ECL3. We are still seeing these interactions but in an altered manner: in the designed model the peptide was predicted to have a D-Tyr (DCNP1_design D-Tyr-R3) interaction with ECL2 and a Tyr (DCNP1_design Tyr-R1) interaction with ECL3. We found that the residue D-Tyr (R3) interacted with the ECL3 versus the predicted interaction with ECL2 (SI Fig. S3).' The authors claim that based on the computational design, they only selected to synthesize 6 compounds but end up with a potent KOR agonist DNCP-β-NalA(1). However, the docking pose of DNCP1 looked very different from the cryo-EM structure. Is the selection of the six compounds pure luck?	Even though we see a noticeable “flip” in the macrocycle between the design and the experimental structure this is not purely luck that these designs were selected. Macrocycles have traditionally been quite difficult to model with high accuracy (Hosseinzadeh et al., Nat Commun, 2021; Mulligan et al., 2021, PNAS). Ultimately this protocol has room for optimization but sets a foundation for how to generate novel molecular tools. Optimization in the later stage of compound design would include structure prediction of the macrocycle itself to understand the dynamic states or if it converges to a single structure and molecular dynamics of the peptide in the complex.	-
12	Lines 330-331, 'It is interesting that D1383.32 appears to be too far to form the H-bond interaction with DNCP-β-NalA(1) and the D1383.32N mutation caused only an 8-fold loss of potency for DNCP-β-NalA(1), but a 1,000-fold for U50,488'. (1) If the cryo-EM structure data shows DNCP-β-NalA(1) does not form HB with D138, which is considered a critical residue/interaction utilized for the computational design of DNCP-β-NalA(1). How reliable would the results of the computational design be? (2) Fig. S14 is important to support this claim, should be moved to and combined with Fig.5. (3) Does U50,488 bind to KOR in the same binding pocket(s) using same sets of residues as does DNCP-β-NalA, or as shown in cyro-EM structure?	(i) Our refined model shows that the ligand 'nitrogen' can form a weak salt-bridge interaction (3.83 Å) with D138. The MD simulations support that this interaction is dynamic, as the interaction was made in less than 28% of the simulations time. Our functional results also showed that either D138A (17-fold) or D138N (7.2-fold) could significantly reduce the DNCP-mediated cAMP inhibition, while they both nearly completely kill the agonist activity of endogenous dynorphin A 1-13 (SI Fig. 26 and SI Table S9). This has been addressed in the text: “It is interesting that D138^{3.32} appears to form a weak salt-bridge interaction (3.8 Å) with DNCP-β-NalA(1) and the D138^{3.32}N mutation caused only an 8-fold loss of potency for DNCP-β-NalA(1), but a 1,000-fold for U50,488 (Fig. 5c, SI Fig. S25, SI Fig. S26 and SI Table S9). This is further confirmed by the reduced binding affinity of DNCP-β-NalA in KOR D138A or D138N mutants (SI Fig. 27 and SI Table S10). This is in accordance with the MD simulations where the distance between the ligand and D138^{3.32} sampled a wide distance distribution (SI Fig. S28).” (ii) In addition, we have combined the D138N of Fig. S14 (old) with Fig. 5c; for clarity we kept SI Fig. 25 (new). (iii) U50,488, as well as pentazocine were docked into the structure of KOR-DNCP-β-NalA(1) using the Schrodinger program. MP1104 was adopted from previously reported crystal structure of KOR-MP1104 (PDB ID: 6B73). Binding of U50,488 in the same pocket as the small-molecule portion of DNCP-β-NalA(1) is further supported by U50,488's close analogue, GR89,696. We have recently reported the cryo-EM structure of KOR-GR89,696-Gz complex (PDB ID: 8DZS), and GR89,696 binds to the orthosteric pocket of KOR similar to MP1104 and β-NalA. This has been addressed in the text: “The small molecule portion of DNCP-β-NalA(1) (bottom half) adopts a conformation similar to MP1104, as well as other typical KOR agonists including U50,488 and pentazocine (SI Fig. S20).”	p. 15 (lines 562-567) p. S28 p. S41 p. 14 p. 13 (line 522ff)
13	Lines 338-340, 'Given that pentazocine is a partial agonist at KOR (Emax: 40% of U50,488) and adopts a similar binding pose as the small-molecule portion of DNCP-β-NalA(1), this suggests that specific residues in the orthosteric pocket are sufficient to	We have modified the main text to “The I135^{3.29}A and K227^{5.39}A appear to specifically affect DNCP-β-NalA(1)'s efficacy in receptor activation (G_{i1} activation, KOR-WT 86%, I135^{3.29}A 46%, K227^{5.39}A 48%), while not or slightly reduces the potency (Fig. 5c and SI Table S11).”	p. 15 (lines 574-576) p. S27-29

	regulate ligand efficacy, as supported by I1353.29A and K2275.39A that cause a loss of 50% efficacy compared to the wild type'. It is not clear what the authors want to claim here. Is interaction with D138 critical or not that important in terms of receptor binding and ligand function?	KOR D138 is known as a critical anchoring residue for ligand binding and agonism, including endogenous dynorphins and small-molecule U50,488 and GR89,696. Our functional characterization shows that either D138A or D138N mutation could significantly reduce DNCP- β -NalA (1)'s potency (Fig. 5c, SI Fig. S25/26) and binding by 10-fold at KOR (SI Fig. S27), whereas they both abolish the activity of dynorphin A 1-13 or U50,488. This difference is likely due to the different interactions formed by the other portions of the ligand which can compensate the loss of interaction with D138. This compensation effect is also observed from MP1104, in which it maintains potent agonist activity in D138A or D138N (Che et al., Cell 2018). Although not related to this study, other exceptions that D138 is not that important for ligand binding or function come from the salvinorins, such as salvinorin A (SaIA) or salvinorin B. Previous study by Vardy et al., J Biol Chem 2013 and our recent work by Han et al., Nature 2023 showed that D138A did not significantly affect the EC50 of SaIA-mediated-cAMP inhibition, whereas D138N further increased the potency by introducing potential hydrogen-bond interactions.	
14	Lines 344-345, 'Several residue mutations, such as E209ECL2A, E2976.58A, and L3097.32A led to increased potency in G protein activation and enhanced efficacy in arrestin recruitment.' Do these residues E209, E297, and L309 interact with DNCP- β -NalA(1) in the cryo-EM structure? How can this information be translated to guide the computational design of a more potent KOR agonist?	Indeed, we mapped out the potential interaction between the cyclic peptide and KOR residues (SI Fig. S19). It suggests that there are potential interactions between DNCP- β -NalA(1) and E209 ^{ECL2} , E297 ^{6.58} , and L309 ^{7.32} as the distances between them are all less than 4.5 Å. The MD simulations also support those potential interactions, although weak, are formed between ligand and E209 ^{ECL2} , E297 ^{6.58} , and L309 ^{7.32} (SI Fig. S28). It is indeed interesting to see that mutations of several residues in the ECL2/3 can further increase the potency and efficacy of DNCP- β -NalA(1). As the extracellular portions of GPCRs undergo dynamic conformational displacement upon receptor activation. We argue that the gain of functions are likely due to the removal of steric clashes when mutating those residues. Consistent with recent reported structures of opioid receptors bound to different endogenous peptides (Wang et al., Cell 2022), our structure and functional results suggest that ECL2/3 and extracellular transmembrane ends play important roles in regulating the efficacy of opioid receptor activation. Future ligand design should consider targeting these less conserved regions to achieve ligand selectivity and efficacy. This has been addressed in the text: "We are still seeing these interactions but in an altered manner: in the designed model the peptide was predicted to have a D-Tyr (DNCP1_design D-Tyr-R3) interaction with ECL2 and a Tyr (DNCP1_design Tyr-R1) interaction with ECL3. We found that the residue D-Tyr (R3) interacted with the ECL3 versus the predicted interaction with ECL2 (SI Fig. S21). Further structural information of the other computationally designed peptides whose ECL2/3 interactions are similar to those of computational DNCP-β-NalA(1) (SI Fig. S22) would elucidate the peptides' ability to induce an altered conformation upon binding." and "The MD simulations indicate potential interactions between the ligand and amino acid residues (E209^{ECL2}, E297^{6.52} and L309^{7.32}). In most simulations, the highest probability of distance consistently remained below 6 Å. However, in certain simulations, larger distance probability values were observed, indicating increased flexibility between the ligand and those specific residues (SI Fig. S28)."	p. S21 p. 13 and p.15 p. S30
15	Lines 366-369, 'we overcome multiple rounds of structure-based design and pharmacological testing; we needed to synthesize and experimentally characterize only 4 compounds to discover a high-affinity molecule with	Our MD simulations (see above) data together with functional data of DNCP- β -NalA(1) at provide additional evidence that de novo designed conjugate is not purely coincidental.	-

	novel patterns of GPCR signaling and pharmacology.' As the docking pose is not validated by cryo-EM structure, it seems the authors are lucky to discover DNCP- β -NalA(1) with only synthesis of 4 compounds. Also, since DNCP- β -NalA(1) exhibited binding at MOR as potent as at KOR, and the functional activity at MOR is not characterized, its GPCR signaling and pharmacology are not well characterized.		
16	Lines 100, 'The internal cycle reduces flexibility and thus reduces the entropy loss upon binding which can increase binding affinity and enhance stability ^{4,28} .' What does 'the internal cycle' mean?	This is a cyclic motif embedded within the peptide sequence. We have reworded the text to make this clear: " The cyclic motif embedded within the peptide sequence reduces flexibility and thus reduces the entropy loss upon binding which can increase binding affinity and enhance stability ".	p. 4 (lines 124-125)
17	Line 103, 'we chose to employ the cyclic component of the lariat 5-6 residues closed by thioether macrocyclization linking a Cys side chain and the N-terminus'. Add (Fig. 1b) to the end of the sentence.	As suggested, reference to Fig. 1b was added to the end of the sentence.	p. 4
18	In Fig 1b (3), what is the structure showing in ball & stick style?	In Fig. 1b (3) a randomly selected set of 6-mer thioether backbones from a comprehensively sampled set of 6mer thioether backbones is shown. The representations here depicts the diversity in backbone shape found in the macrocycles.	p. 5
19	SI Fig S8, add calc. [M+H] ⁺ to each structure.	This has been done as suggested. The calculated monoisotopic mass [M+H] ⁺ has been added to each spectrum (SI Fig. S8).	p. S9-10

Reviewers' Comments:

Reviewer #1:

Remarks to the Author:

All suggested changes have been made.

Reviewer #2:

Remarks to the Author:

The addition of CFA data is certainly helpful, but hardly what I was actually asking for. Pain three days after an inflammatory injection is not "chronic". What I was hoping to see was von Frey allodynia weeks (or better, months) after some experimental nerve damage assay. This would also address the issue of whether DNCP- β -NalA(1) would be predicted to be efficacious against neuropathic pain in addition to inflammatory pain. I'm not sure whether to insist on this, but it would *greatly* strengthen the prediction of human efficacy here.

The authors have provided possibly the most infuriating response possible to my complaint about their use of only male animals. They say they agree, and that they "considered" it, but then trot out the oldest excuse in the book, and one that has been thoroughly debunked. It is NOT the case that female animals have more variability because of their estrous cycle (and it's an estrous cycle in mice, not a menstrual cycle) (e.g., Prendergast et al., NBR, 2014), and it is NOT the case that female gonadal hormones affect the results of pain experiments under normal circumstances (see Mogil & Chanda, Pain, 2005). The authors are correct that the use of only male animals has been "common practice", but it is also the case that this common practice has been strongly and appropriately criticized (e.g., Shansky, Science, 2019). It would hardly be onerous for the authors to repeat their behavioral experiments to allow them to predict that this compound would be efficacious in half the population, and the clear majority of pain patients. This is especially true since a literature already exists documenting sex differences in the efficacy of kappa-opioid analgesia (e.g., Gear et al., Nat. Med., 1996).

Reviewer #3:

Remarks to the Author:

I appreciate the effort of the authors on improving the structure, as well as the additional MD and pharmacological data that have addressed most of my concerns. However:

1. The MD simulation address some issues but also leads a bit more problems as follows:

1a. Why set 6 Å cut-off as threshold for contact definition? That is unusually large. I wonder if 4 Å or 5 Å has similar tendency. Also, contact tendency is a general evaluation, if possible, the author should analyze the detailed interaction type, such as hydrogen bond, salt bridge or hydrophobic interactions.

1b. The method of simulations needs to be clarified. Firstly, is Amber23 a correct version number? Currently developed version should be named as AmberTools23 and Amber22. Secondly, the reasonability of membrane component should be verified, has such component been used in other publications? Thirdly, Amber99SB has been out-of-date, the authors should prove the force field combination (99SB with lipid21) will not cause problem in simulations. Fourthly, the parameters of Langevin thermostat and the Monte Carlo barostat should be claimed for reproductivity.

2. Since the cyclic peptide moiety of DNCP- β -NalA(1) has incomplete density in the map, much likely due to the unstable interaction and potential steric clashes of the cyclic part with KOR extracellular vestibule. The additional MD data are helpful to verify the possibility of the contact of the cyclic peptide moiety with ECL2, ECL3, and extracellular sides of TM6/7. However, the detailed large hydrophilic interaction network shall not be induced as shown in Fig.S19, which is controversial. Instead, I would suggest the authors to remove Fig. S19 and directly use the MD data of the cyclic peptide of DNCP- β -NalA(1) and residues in the KOR extracellular vestibule to

characterize the contact profiles of DNCP- β -NalA(1) with the ECL2, ECL3, and TM6/7.

3. In Fig.S17a, the molecular weight of KOR shown in the gel is below 25kDa, which is too small for a GPCR. As the author suggested, the engineered KOR construct used in this study contains more than 300 amino acids. That is to say, the molecular weight of KOR should be more than 30kDa. Did the author used the wrong image data?

Reviewer #4:

Remarks to the Author:

The research is significant to the drug discovery field aiming to develop novel opioid ligands with improved pharmacological profile. The approach for generating cyclic peptide-small molecule conjugates is novel; the methodology is sound, and the results support their conclusions.

My questions and concerns have been addressed. I recommend it for publication in Nature Communications.

#	Comments	Response	Page #
Reviewer 1			
1	All suggested changes have been made.	Noted with gratitude.	
Reviewer 2			
1	The addition of CFA data is certainly helpful, but hardly what I was actually asking for. Pain three days after an inflammatory injection is not "chronic". What I was hoping to see was von Frey allodynia weeks (or better, months) after some experimental nerve damage assay. This would also address the issue of whether DNCP-beta-NalA(1) would be predicted to be efficacious against neuropathic pain in addition to inflammatory pain. I'm not sure whether to insist on this, but it would *greatly* strengthen the prediction of human efficacy here.	We thank reviewer for the comments and apologize for the imprecise use of the term "chronic" pain in the previous of the manuscript. While we agree that an additional long-term pain model (such as the neuropathic pain model with von Frey allodynia) would further strengthen and increase the translational value of our work, we have now carefully revised the text: (a) the animal pain models we used were referred to by name, and (b) we avoided a direct linking of chronic pain with the animal model data.	abstract page 2; p. 10-12; p. 27/28
2	The authors have provided possibly the most infuriating response possible to my complaint about their use of only male animals. They say they agree, and that they "considered" it, but then trot out the oldest excuse in the book, and one that has been thoroughly debunked. It is NOT the case that female animals have more variability because of their estrous cycle (and it's an estrous cycle in mice, not a menstrual cycle) (e.g., Prendergast et al., NBR, 2014), and it is NOT the case that female gonadal hormones affect the results of pain experiments under normal circumstances (see Mogil & Chanda, Pain, 2005). The authors are correct that the use of only male animals has been "common practice", but it is also the case that this common practice has been strongly and appropriately criticized (e.g., Shansky, Science, 2019). It would hardly be onerous for the authors to repeat their behavioral experiments to allow them to predict that this compound would be efficacious in half the population, and the clear majority of pain patients. This is especially true since a literature already exists documenting sex differences in the efficacy of kappa-opioid analgesia (e.g., Gear et al., Nat. Med., 1996).	We apologize for triggering such anger and disharmony with our previous response. Clearly, this was not our intention. We must admit that we have never been criticized in this matter previously, and simply were not sensitized appropriately in this matter, although we should have known better. Your constructive points have been well taken and are much appreciated. Currently, the performance of these additional experiments with females is limited by ethical aspects. An animal license is necessary for performing the behavioral studies in females, which would require several months for approval by the respective authorities. However, we will implement this in our future research. While we agree that additional data in pain assays using female mice would be important to increase the translational value of our study, we argue that the focus of our study is on the computer-assisted de novo design and synthesis of a GPCR ligand, and its structural validation, while the animal experiments were performed to demonstrate proof of concept for efficacy, which has been exemplarily demonstrated in males. Therefore, given all the above said, we find it appropriate and scientifically accurate to modify the text of our manuscript (throughout): we transparently highlighted that animal experiments and efficacy data were collected in male mice only.	abstract page 2; p. 10-12; p. 20; p. 27/28
Reviewer 3			
1	I appreciate the effort of the authors on improving the structure, as well as the additional MD and pharmacological data that have addressed most of my concerns. However:	Thank you. Much appreciated.	
2	1. The MD simulation address some issues but also leads a bit more problems as follows: 1a. Why set 6 Å cut-off as threshold for contact definition? That is unusually large. I wonder if 4 Å	Initially, we set a 6 Å cut-off for the contact definition to accommodate the potential flexibility of amino acid side chains. This choice was made because the contact distance between the ligand and amino acid residue can vary as the side chains fluctuate.	page 13; p. 25-26; p. S26

	or 5 Å has similar tendency. Also, contact tendency is a general evaluation, if possible, the author should analyze the detailed interaction type, such as hydrogen bond, salt bridge or hydrophobic interactions.	When we altered the contact definition to 5 Å or 4 Å, we observed a similar trend, but the proportion of contact values decreased as the cut-off distance decreased (see below, Figure for review only). This decrease was attributable to the dynamic nature of both the ligand and amino acid side chains. However, to ensure clarity, we now used a cut-off distance of 4.5 Å in SI Figure S24. We now highlighted this by using different colours in the figure to distinguish between hydrophobic contacts and hydrogen bonding interactions.  Figure: Fraction of contact observed in the combined 2 μs trajectories of molecular dynamics simulations. A cut-off distance of (a) 5 Å and (b) 4.5 Å (c) 4.0 Å between all heavy atoms of DNCP-β-NaIA(1) and KOR are used for contact analysis.	
3	1b. The method of simulations needs to be clarified. Firstly, is Amber23 a correct version number? Currently developed version should be named as AmberTools23 and Amber22. Secondly, the reasonability of membrane component should be verified, has such component been used in other publications? Thirdly, Amber99SB has been out-of-date, the authors should prove the force field combination (99SB with lipid21) will not cause problem in simulations. Fourthly, the parameters of Langevin thermostat and the Monte Carlo barostat should be claimed for reproductivity.	We thank the reviewer for this valuable comment. We have thoroughly revised and corrected the methods section and updated it. We have used software Amber22 and AmberTools23. We have used Amber19SB with LIPID21 forcefield. The same membrane component was used in the following publication (Han, J. et al., Nature 2023). The methods section was updated with Langevin thermostat and Monte Carlo barostat parameters.	
4	2. Since the cyclic peptide moiety of DNCP-β-NaIA(1) has incomplete density in the map, much likely due to the unstable interaction and potential steric clashes of the cyclic part with KOR extracellular vestibule. The additional MD data are helpful to verify the possibility of the contact of the cyclic peptide moiety with ECL2, ECL3, and extracellular sides of TM6/7. However, the detailed large hydrophilic interaction network shall not be induced as shown in Fig.S19, which is controversial. Instead, I would suggest the authors	We thank the reviewer for this suggestion. We replaced previous SI Figure S19 with MD data (SI Figure S23).	p. 25/26; p. S25

	to remove Fig. S19 and directly use the MD data of the cyclic peptide of DNCP-β-NalA(1) and residues in the KOR extracellular vestibule to characterize the contact profiles of DNCP-β-NalA(1) with the ECL2, ECL3, and TM6/7.		
5	3. In Fig.S17a, the molecular weight of KOR shown in the gel is below 25kDa, which is too small for a GPCR. As the author suggested, the engineered KOR construct used in this study contains more than 300 amino acids. That is to say, the molecular weight of KOR should be more than 30kDa. Did the author used the wrong image data?	We thank the reviewer for pointing this out. The inconsistency between gel position and molecular weight appears to be common for membrane proteins due to the gel shifting effect associated with factors such as their shape, radius and charge alterations following running buffer and SDS molecules binding, which has been extensively studied in Rath A et al., PNAS 2009. To support our observation, here we conducted SDS-PAGE analysis including three samples, the uncleaved KOR, cleaved KOR by 3C protease, and 3C protease alone. As shown below in Figure 1A (for revision only), the uncleaved KOR (Bril-KOR, MW=52 kDa) also showed a lower position near 40 kDa; the cleaved KOR (MW=34 kDa) used in this study exhibits a lower migration pattern, approximately below 25 kDa. As explained by the gel shifting effect, the KOR proteins may migrate much faster than anticipated likely due to an increased net negative charge with the running buffer or a more compact structure because of the binding of SDS molecules. The similar effect has also been observed in our earlier work on 5HT_{2A}R as shown in Figure 1B (Kim K et al., Cell 2020).  Figure. The SDS-PAGE analysis of uncleaved and cleaved KOR proteins. A, Lane 1: the uncleaved KOR proteins; Lane 2: the cleaved KOR proteins; Lane 3: the 3C protease. B, The cleaved 5HT_{2A}R displayed a SDS-PAGE position near 25 kDa. Figure 1B is adopted from Kim K et al., Cell 2020.	

Reviewer 4			
1	The research is significant to the drug discovery field aiming to develop novel opioid ligands with improved pharmacological profile. The approach for generating cyclic peptide-small molecule conjugates is novel; the methodology is sound, and the results support their conclusions. My questions and concerns have been addressed. I recommend it for publication in Nature Communications.	Thank you very much!